



# Methane chemistry in a nutshell – The new submodels CH4 (v1.0) and TRSYNC (v1.0) in MESSy (v2.54.0)

Franziska Winterstein[1] and Patrick Jöckel[1]

[1]Deutsches Zentrum für Luft- und Raumfahrt (DLR), Institut für Physik der Atmosphäre, Oberpfaffenhofen, Germany

**Correspondence:** Franziska Winterstein (franziska.winterstein@dlr.de)

**Abstract.** Climate projections including chemical feedbacks rely on state-of-the-art chemistry-climate models (CCMs). Of particular importance is the role of methane ($CH_4$) for the budget of stratospheric water vapor (SWV), which has an important climate impact. However, simulations with CCMs are, due to the large number of involved chemical species, computationally demanding, which limits the simulation of sensitivity studies.

5  To allow for sensitivity studies and ensemble simulations with a reduced demand for computational resources, we introduce a simplified approach to simulate the core of methane chemistry in form of the new Modular Earth Submodel System (MESSy) submodel CH4. It involves an atmospheric chemistry mechanism reduced to the sink reactions of $CH_4$ with predefined fields of the hydroxyl radical (OH), excited oxygen ($O(^1D)$), and chlorine (Cl), as well as photolysis and the reaction products limited to water vapour ($H_2O$). This chemical production of $H_2O$ is optionally feed back onto the specific humidity (q) of the connected

10  General Circulation Model (GCM), to account for the impact onto SWV and its effect on radiation and stratospheric dynamics.

The submodel CH4 is further capable of simulating the four most prevalent $CH_4$ isotopologues for carbon and hydrogen ($CH_4$ and $CH_3D$ as well as $^{12}CH_4$ and $^{13}CH_4$), respectively. Furthermore, the production of deuterated water vapour (HDO) is, similar to the production of $H_2O$ in the $CH_4$ oxidation, optionally feed back to the isotopological hydrological cycle simulated by the submodel H2OISO, using the newly developed auxiliary submodel TRSYNC. Moreover, the simulation of a user defined

15  number of diagnostic $CH_4$ age- and emission classes is possible, which output can be used for offline inverse optimization techniques.

The presented approach combines the most important chemical hydrological feedback including the isotopic signatures with the advantages concerning the computational simplicity of a GCM, in comparison to a full featured CCM.

## 1 Introduction

It is beyond question that methane ($CH_4$) is a strong greenhouse gas (GHG), with an estimated global warming potential (GWP) of 34 times that of carbon dioxide ($CO_2$) on a 100 year horizon (IPCC, 2013). Therefore, most General Circulation Models (GCMs) include the effect of the radiative forcing of $CH_4$. However, the effect of $CH_4$ is underrepresented by only





using its direct radiative impact and not accounting for the water vapour ($H_2O$) produced by the oxidation of $CH_4$ due to a set-up without chemistry. Especially in the stratosphere this additional $H_2O$ (stratospheric water vapor (SWV)) influences among others the radiative forcing, stratospheric temperature and the ozone ($O_3$) chemistry (Stenke and Grewe, 2005; Tian et al., 2009; Solomon et al., 2010; Revell et al., 2012; Winterstein et al., 2019). The inclusion of production of $H_2O$ by $CH_4$ requires a chemical mechanism as provided by chemistry-climate models (CCMs). Current state-of-the-art CCMs include a vast amount of chemical species and reactions. By extending the chemical mechanisms, one intends to achieve an increase in accuracy of the atmospheric chemistry representation. At the same time, however, the computational demands increase. Although, available computational power increases at a certain rate, too, the availability and capacity of high performance computers is a limiting factor for sensitivity studies in climate projection simulations with CCMs.

It is hence advisable to recognize both main effects of $CH_4$, namely its radiative forcing and its impact on SWV, but keeping computational demands low at the same time. Therefore, our approach to simulate $CH_4$ includes both effects and is able to use predefined reaction partners of $CH_4$, which reduces computational cost to a minimum.

Sections 1.1 and 1.2 introduce the sources and sinks of $CH_4$, and $CH_4$ isotopologues and their fractionation effects, respectively. In Sect. 2 we briefly present the Modular Earth Submodel System and describe the concept of the CH4 submodel in Sect. 3. Two additional options of the CH4 submodel are explained in the subsequent Sects. 3.1 and 3.2. The coupling to the hydrological cycle with the submodel TRSYNC is introduced in Sect. 4. We show three example applications using the newly presented submodels in Sect. 5 and end with a short summary. Parts of the manuscript are based on the PhD thesis of the first author (Frank, 2018).

### 1.1 Sources and sinks of $CH_4$

Methane is a GHG emitted by both, natural, and anthropogenic sources at the Earth's surface. There are basically no known chemical sources of $CH_4$ in the free atmosphere.

In CCMs usually predefined lower boundary conditions instead of emission fluxes are used to describe atmospheric $CH_4$. This approach is mainly employed due to two major problems: (1) The simulated $CH_4$ lifetime is not sufficiently accurate, however important for tropospheric and stratospheric chemistry. Thus, realistic climate projections with interactive chemistry and $CH_4$ emission fluxes are difficult. (2) Despite large ongoing efforts, current emission inventories are still subject to large uncertainties, as top-down and bottom-up inventories differ significantly (e.g. EDGAR or Saunois et al. (2016)). This mismatch indicates the dilemma, that there are a lot of open questions with respect to both, the magnitude of sources, and the sinks of $CH_4$.

Methane is removed from the atmosphere mainly by three photochemical reactions and is also depleted by photolysis:

$$\frac{d[CH_4]}{dt} = (-k_{CH_4+OH} \cdot c_{\text{air}} \cdot [OH] - k_{CH_4+O(^1D)} \cdot c_{\text{air}} \cdot [O(^1D)] - k_{CH_4+Cl} \cdot c_{\text{air}} \cdot [Cl] - p_{CH_4+h\nu}) \cdot [CH_4], \qquad (1)$$

where [X] denotes the mixing ratio of species X in [mol mol$^{-1}$], $c_{\text{air}}$ the concentration of air in [cm$^{-3}$], $k_R$ the reaction rate coefficient of reaction $R$ in [cm$^3$ s$^{-1}$] and $p_{CH_4+h\nu}$ the photolysis rate of $CH_4$ in [s$^{-1}$].





About 88 % of the atmospheric $CH_4$ removal happens in the troposphere. The largest part is thereby the reaction with the hydroxyl radical (OH) (95 % of the tropospheric sink), while the rest is attributed to the reaction with chlorine (Cl) in the Marine Boundary Layer (MBL). About 8 % of $CH_4$ is depleted in the stratosphere, by the reactions with OH, excited oxygen $(O(^1D))$, Cl and through photolysis (IPCC, 2013).

Another sink of $CH_4$ is the so called soil-loss at the Earths' surface. $CH_4$ is either depleted by $CH_4$ consuming bacteria (methanotrophs), or it is removed from the air by diffusive transport into the soil, which is mostly influenced by soil water content (King, 1997). Globally, the soil-loss accounts for approximately 4 % of the total $CH_4$ sink (IPCC, 2013).

## 1.2   Isotopologues of $CH_4$

A powerful and common method in the investigation of the $CH_4$ budget is the study of $CH_4$ isotopologues. Production and
removal of $CH_4$ cause fractionation effects, which lead to distinct isotopological signals in the atmosphere. These isotopic signatures provide potentially additional insights into the role of specific $CH_4$ sources and depleting reactions, and are already widely used in the context of $CH_4$ (Hein et al., 1997; Fletcher et al., 2004; Monteil et al., 2011; Rigby et al., 2012; Nisbet et al., 2016; Schaefer et al., 2016).

Fundamentally, the stable isotopologues of $CH_4$ form with respect to the most abundant stable isotopes of hydrogen and of
carbon. The stable isotopes of hydrogen are $^1H$ and $^2H$ (deuterium, D), and for carbon, carbon-12 ($^{12}C$) and carbon-13 ($^{13}C$). This results in the first order stable isotopologues $^{12}CH_4$, $^{13}CH_4$, and $CH_3D$. The relative abundances of higher substituted and mixed isotopologues (e.g. $CH_2D_2$ or $^{13}CH_3D$) are less than $0.0007\%$ (compared to $0.0616\%$ of $CH_3D$) (Stolper et al., 2014) and hence neglected.

The chemical fractionation is based on the fact that isotopologues of the same molecule have different reaction rates, i.e.
they react with different speed or probability. This difference in reaction rates is described as the so called Kinetic Isotope Effect (KIE) and becomes apparent during the chemical reaction of a specific molecule X:

$$X_L + E \quad \overset{k_L}{\rightarrow} P \tag{R1}$$
$$X_H + E \quad \overset{k_H}{\rightarrow} P' \tag{R2}$$

with $X_L$ being its light (major), and $X_H$ its heavy (minor) isotopologue. E and P/P' denote the reaction partner(s) and prod-
uct(s), respectively. The value of the KIE is thereby defined as the ratio of the reaction rates $k_L$ and $k_H$ (Bigeleisen, 2005) and its inverse is called the fractionation factor $\alpha$:

$$\text{KIE} := \frac{k_L}{k_H} = \frac{1}{\alpha} \; . \tag{2}$$

The KIEs of the sink reactions of $CH_4$ have been, among others, determined by Saueressig et al. (1995, 1996, 2001) and Crowley et al. (1999) in laboratory measurements (see Table 1). Since the KIEs of $CH_4$ isotopologues are partly temperature
dependent, the KIEs are described by two parameters A and B and are calculated as

$$\text{KIE} = \text{A} \cdot \exp(\text{B}/T), \tag{3}$$



**Table 1.** Temperature dependent KIEs of the sink reactions of $CH_4$ described as $KIE = A \cdot \exp(B/T)$. The KIEs are valid in the given temperature range (T in [K]).

| reaction | A | B | T | reference |
|---|---|---|---|---|
| $KIE^{OH}_{^{13}CH_4}$ | 1.0039 | 0.0 | 200–300 | (Saueressig et al., 2001) |
| $KIE^{O(^1D)}_{^{13}CH_4}$ | 1.013 | 0.0 | 223–295 | (Saueressig et al., 2001) |
| $KIE^{Cl}_{^{13}CH_4}$ | 1.043 | 6.455 | 223–297 | (Saueressig et al., 1995; Crowley et al., 1999) |
| $KIE^{OH}_{CH_3D}$ | 1.097 | 49.0 | 249–422 | (Saueressig et al., 2001) |
| $KIE^{O(^1D)}_{CH_3D}$ | 1.060 | 0.0 | 224–295 | (Saueressig et al., 2001) |
| $KIE^{Cl}_{CH_3D}$ | 1.278 | 51.31 | 223–295 | (Saueressig et al., 1996) |

with $T$ being the temperature in [K].

The largest KIE and therefore strongest fractionation effect is found for the reaction with Cl, which especially influences the isotopic composition of $CH_4$ in the middle and upper stratosphere (Saueressig et al., 1996; Bergamaschi et al., 1996).

Conversely, the reaction with $O(^1D)$ shows the lowest KIE, which furthermore does not show any temperature dependence. The KIE of the reaction with OH is temperature dependent with respect to deuterated methane ($CH_3D$) but not with respect to methane containing $^{13}C$ ($^{13}CH_4$) (Saueressig et al., 2001). Nair et al. (2005) estimated the rate coefficients of the photodissociation of $CH_4$ and its major isotopologues for planet Mars, which results in a calculated KIE= 1.005 for $CH_3D$ and a negligible isotopic fractionation for the $^{13}C$ isotopologue (Nixon et al., 2012). There is, especially for deuterium, a non-negligible frac-

tionation during the soil-loss for $CH_4$ (Snover and Quay, 2000; Maxfield et al., 2008). An average value for the overall soil-loss is estimated as $KIE^{soil}_{CH_3D} = 1.0825$ and $KIE^{soil}_{^{13}CH_4} = 1.0196$ (Snover and Quay, 2000; Holmgren, 2006; Maxfield et al., 2008).

## 2    The Modular Earth Submodel System (MESSy)

The framework of the Modular Earth Submodel System (MESSy, used in the second version MESSy2, Jöckel et al. (2010)) is based on the idea to modularize a climate model in a way, that single components can be switched on and off independently,

depending on the desired set-up, meeting the demands of current Earth System Modeling in terms of flexibility and computational performance. The modularization enables the user to pick suitable submodels or expand the model easily with new ones. The here presented new submodel for simplified $CH_4$ chemistry (CH4) and the auxiliary submodel TRacer SYNCHronization (TRSYNC) are implemented based on this framework.

For the application examples of the new submodel, MESSy is used together with the core atmospheric model 5th generation

European Centre Hamburg general circulation model (ECHAM5, Roeckner et al. (2006)). The ECHAM/MESSy Atmospheric Chemistry (EMAC) model is a numerical chemistry and climate simulation system that includes sub-models describing tropospheric and middle atmosphere processes and their interaction with oceans, land and human influences (Jöckel et al., 2010). EMAC (ECHAM5 version 5.3.02, MESSy version 2.54, Jöckel et al. (2010, 2016)) is applied in the given examples in the





T42L90MA-resolution, i.e. with a spherical truncation of T42, which corresponds to a quadratic Gaussian grid of approx. 2.8
by 2.8 degrees in latitude and longitude, and includes 90 vertical hybrid pressure levels from the Earth surface up to 0.01 hPa.
MESSy allows the configuration of EMAC in several operational modes. The two basic ones are the GCM set-up without
chemistry and the CCM set-up with fully interactive chemistry, using, among other components, the Module Efficiently Cal-
culating the Chemistry of the Atmosphere (MECCA, Sander et al. (2005)). MECCA and the SCAVenging (SCAV, Tost et al.
(2006)) submodel represent the chemical kinetics of EMAC in gas phase and in aqueous phase, respectively. They define the
underlying chemical reaction mechanisms in troposphere, stratosphere, and lower mesosphere. MECCA and SCAV provide
comprehensive mechanisms, combining state-of-the-art reactions and rate coefficients. The kinetic chemistry tagging technique
(MECCA_TAG, Gromov et al. (2010)) enables the user to tag selected chemical elements, without modifying the underlying
standard chemical mechanism of MECCA. It can be applied for simulating isotopologues of trace gases with respect to selected
isotopes. In order to do so, rare and abundant isotopologues of the species of interest (e.g., those containing atomic hydrogen
(H)) are created in an extended set of reactions in the same chemical mechanism.

MESSy and its application in EMAC has been used in multiple studies (see the special issue in Atmospheric Chemistry
and Physics https://www.atmos-chem-phys.net/special_issue22.html) and includes several submodels from contributing insti-
tutions. Further information on EMAC, MESSy and its submodels can be found in Jöckel et al. (2010, 2016), on the web-site
https://www.messy-interface.org/, or accompanying papers documenting the specific submodels.

## 3  The submodel CH4

The MESSy submodel CH4 aims to close the gap between the operational modes of EMAC as a GCM without chemistry
and as a CCM with the comprehensive chemical mechanisms of MECCA and SCAV. The basic concept of the submodel
is to limit the chemical mechanism to the loss-processes of methane and use predefined fields of the reaction partners OH,
$O(^1D)$ and Cl to reduce the computational demands. The predefined fields are taken either from existing simulation results
with detailed chemistry, or from other data sources (e.g. reanalyses or projections). If CH4 is included in an EMAC CCM
simulation (which is possible in the MESSy framework), the CH4 submodel can also be coupled to the reactant fields, which
are on-line calculated during the same simulation by the chemical mechanism (i.e. MECCA). Although this does not save
computational requirements, such a simulation configuration can be used, for example, if output of one of the additional
options of the CH4 submodel are desired together with a coupled comprehensive chemical mechanism. Same applies for the
photolysis rate of $CH_4$, which can be predefined or on-line calculated by the submodel JVAL (Sander et al., 2014).

Figure 1 visualizes the conceptual differences between the MESSy submodel CH4 (left) and a CCM simulation with
MECCA (right). MECCA simulates the entire chemical mechanism and therefore also includes the feedback onto the reaction
partners (depicted in yellow) of $CH_4$. Additionally, there is also a secondary feedback by the products from the $CH_4$ sink
reactions (e. g. $H_2O$, $HO_2$, depicted in blue). Conversely, the CH4 submodel uses the predefined fields of the reactant species
to calculate the $CH_4$ loss. This loss is included in the master tracer of the CH4 submodel, but does not feedback onto the sink
fields or any other chemical species, except $H_2O$, in the case when the hydrological feedback of $CH_4$ oxidation is switched



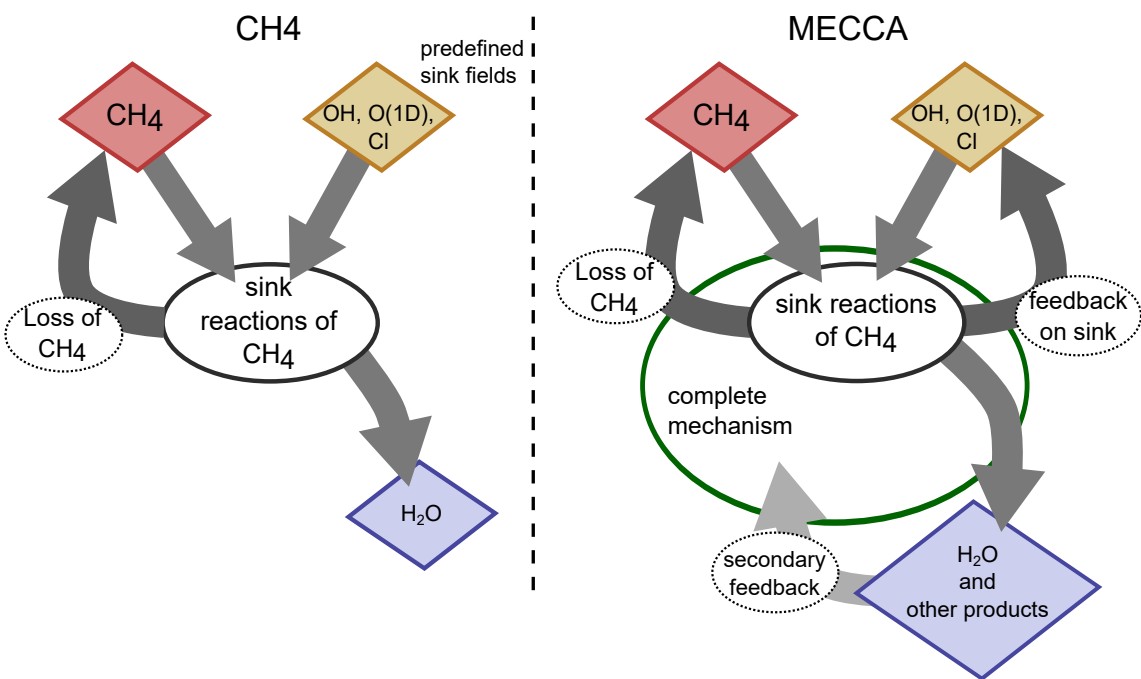

**Figure 1.** Sketch visualizing the concepts of parameterizing $CH_4$ sink reactions in the MESSy submodels CH4 (left) and MECCA (right). The chemical mechanism in CH4 is reduced to the sink reactions of $CH_4$ and gives optional feedback to $H_2O$ only. In MECCA a complete chemical mechanism is included which feeds back among others on $H_2O$ and other products of the $CH_4$ sink reactions. Reaction partners are depicted in yellow, whose feedback is included in MECCA. The reaction partners in the CH4 submodel are predefined fields without feedback.

on. GCMs include $CH_4$ foremost for its radiative impact as a greenhouse gas, but also for its influence on stratospheric water vapor (SWV, e.g. Monge-Sanz et al. (2013); ECMWF (2007); Austin et al. (2007); Boville et al. (2001); Mote (1995)). The CH4 submodel is likewise equipped with an optional feedback onto $H_2O$, to account for the secondary climate feedback of

$CH_4$. It is thereby assumed that two molecules of $H_2O$ are produced per oxidized $CH_4$ molecule (le Texier et al., 1988), which is, however, only a rough approximation as analyzed by Frank et al. (2018).

Note that soil loss is not explicitly included in the CH4 submodel, since the concept of dry deposition is already part of the EMAC submodel DDEP (Kerkweg et al., 2006a). An example how to use DDEP to simulate the soil loss of $CH_4$ is included in the supplement of this paper.

The submodel CH4, with its four sink reactions of $CH_4$, is considerably computationally cheaper, compared to a fully interactive chemistry simulation using MECCA, which represents (depending on the chosen set-up) several hundred reactions (e.g., more than 300 in the base simulations of Earth System Chemistry integrated Modelling (ESCiMo) project (Jöckel et al., 2016)). For example, a reference set-up with MECCA requires about 250 node-h[1] per simulated year, while a set-up with the

---

[1]node-h: required wall-clock hours times applied high performance computer (HPC) nodes.





CH4 submodel without MECCA requires only 30 node-h per year (these numbers are calculated for simulations conducted on
the high performance computer (HPC) Mistral at the *Deutsches Klimarechenzentrum* (DKRZ)).

First simulations using the CH4 submodel are presented in studies by Eichinger et al. (2015a, b), it was included in the simulations of the ESCiMo project (Jöckel et al., 2016) and it has been used for the $CH_4$ forecast system presented by Nickl et al. (2019).

### 3.1   Option I: Age and Emission classes

The $CH_4$ submodel includes an option for simulating age and emission classes. These classes, which can be specified by the user via namelist, enable a precise distinction between $CH_4$ source sectors and/or regions (emission classes), as well as further insight into the $CH_4$ distribution over time (age classes). The term "emission class" denotes thereby a $CH_4$-like tracer defined by the CH4 submodel. The assignment of specific emission fluxes (sectors and regions) to the tracers of the emission classes is handled by the submodel OFFEMIS (Kerkweg et al., 2006b). In our present application example these classes are subject to
emissions being a combination of an emission sector (like wetlands, biomass burning, anthropogenic etc.) and a region (e.g. continents or countries). One tracer, for example, thus traces anthropogenic $CH_4$ emitted from Africa, as shown in Sect. 5.1. These additional diagnostic tracers are transported identical to the master $CH_4$ tracer of the CH4 submodel and also experience the same sink reactions.

The time period represented by one age class can be chosen by the user. How the age and emission classes evolve over time
is depicted in Fig. 2. Methane of each emission class is propagated through a specific number of age classes. The emitted $CH_4$ of a specific emission class is added to the tracer which corresponds to the first age class. After the selected time span it moves to the next "older" age class until it reaches the oldest. The oldest age class represents the background, since $CH_4$ does not proceed further.

It is further selectable which age evolving method is applied. The CH4 submodel offers three options: (1) $CH_4$ is passed on
in one step after a user-defined time-span, (2) $CH_4$ is continuously passed on with respect to an user-defined time-span, and (3) $CH_4$ is passed on monthly with fixed-lag.

We define the state vector for emission class $i$ and age classes 1 to $N$ as:

$$f_i = \begin{pmatrix} f_{i1} \\ f_{i2} \\ \vdots \\ f_{iN} \end{pmatrix} \tag{4}$$

The first two options are implemented according to

$$\Delta f_i = \frac{M \cdot f_i}{\Delta t}, \tag{5}$$




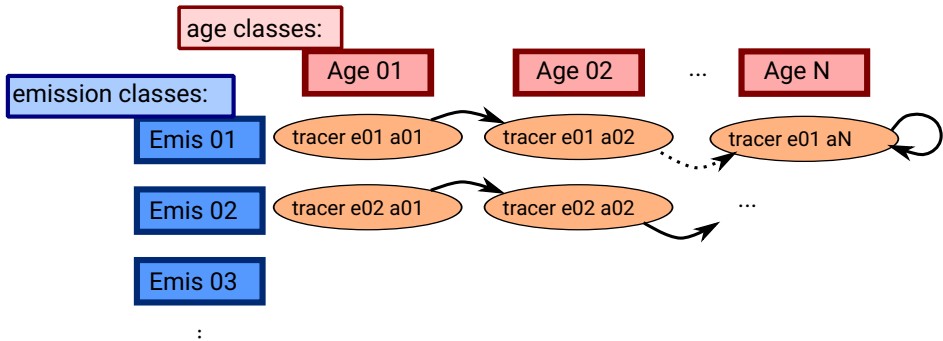

**Figure 2.** Sketch showing the advancing of the age classes in the CH4 submodel. Each tracer represents one specific emission and age class. After the defined length of time, the age classes proceed to the next "older" age class. The last class represents the background $CH_4$, where the $CH_4$ is only subject to transport and the chemically defined sink reactions, but not propagated to an older age class, which is indicated by the circled arrow.

with $\Delta f_i$ being the tendency of $f_i$, $\Delta t$ being the time step length, and $M$ being a matrix defining the ageing step according to the chosen option. For option (1) this matrix looks like

$$
M = \begin{pmatrix}
-1 & 0 & & \cdots & & 0 \\
1 & -1 & & & & \\
& 1 & -1 & & & \vdots \\
& & & \ddots & \ddots & \\
0 & & \cdots & & 1 & 0
\end{pmatrix}.
\tag{6}
$$

This moves the current values of one age class tracer after a user-defined time-span to the next older one. The implementa-
tion of this option is not conform with a Leapfrog time stepping with Asselin-filter and might cause numerical oscillations with negative values. It was implemented solely for testing purposes during development, but it is not recommended for real applications. The ageing step matrix $M$ for option (2) is $M'$

$$
M' = \alpha \cdot M,
\tag{7}
$$

with $\alpha = \frac{\Delta t}{\tilde{T}}$ and $\tilde{T}$ being the user-defined time-span indicating the binning width of the age class. This option carries out a
quasi-continuous update of the age classes, as it moves at every time step a fraction of the current age class to the next.

The third option is implemented for usage by a fixed-lag Kalman filter for inverse optimization. With this option, one age class represents one month and at the end of one month all $CH_4$ of one age class moves to the next. This option is specifically implemented to be conform with the Leapfrog time stepping (c.f. option (1)).

In order to reduce numerical errors, the age and emission classes are continuously constrained (i.e., in each model time step)
to sum up to the master tracer and are scaled appropriately, if the sum deviates.





## 3.2 Option II: Isotopologues

Additional to solving the basic $CH_4$ kinetics, the submodel CH4 further allows for the simulation of $CH_4$ isotopologues, which are a potent diagnostic measure in the source and sink attribution. The submodel CH4 is able to simulate the abundant and first order rare isotopologues and defines these as tracers additional to the master tracer. Higher substituted isotopologues are

neglected. The user can choose, whether isotopologues are simulated with respect to carbon (methane containing $^{12}C$ ($^{12}CH_4$) and $^{13}CH_4$), or hydrogen ($CH_4$ (containing $^1H$ isotopes only) and $CH_3D$), or both. The abundant (with $^{12}C$ or $^1H$ isotopes only) and rare (with $^{13}C$ or D) isotopologues are thereby simulated in parallel. During the simulation it is taken care that each isotopologue family sums up to the master tracer $CH_4$ tracer of the CH4 submodel (`CH4_fx`). The isotopic signatures of $CH_4$ emission sources are included by splitting the emission fluxes into an abundant and a rare fraction. This is handled via the

OFFEMIS namelist (Kerkweg et al. (2006b), see example namelists in the supplement).

The reaction rates of the $CH_4$ isotopologues with their reaction partners are adjusted with respect to the KIE factors, e.g.:

$$KIE = k_{CH_4+OH}/k_{CH_3D+OH} \,, \tag{8}$$

and similar for reactions with $O(^1D)$, Cl, and photolysis. The applied reaction partners are thereby the same as those used for the master tracer.

The oxidation of $CH_3D$ produces to a certain extent deuterated water vapour (HDO). If the feedback of $CH_4$ oxidation onto the hydrological cycle and the simulation of D containing isotopologues is switched on in the CH4 submodel, an additional tracer for HDO is created by the submodel and filled by the produced HDO from $CH_3D$ oxidation. There are two options available: (1) one oxidized $CH_3D$ produces one HDO molecule, or (2) the tendency of the HDO tracer is calculated by Eq. (9) (Eichinger et al., 2015a):

$$\frac{\partial(HDO)}{\partial t} = \frac{-\frac{\partial(CH_3D)}{\partial t} + 6.32 \times 10^{-5} \cdot \frac{\partial(CH_4)}{\partial t}}{\frac{M_{air}}{M_{HDO}} \left(\frac{1}{1-HDO}\right)^2} \,, \tag{9}$$

with $M_{air}$ and $M_{HDO}$ being the molar masses of air (28.987 g mol$^{-1}$) and HDO (19.02 g mol$^{-1}$), respectively. This empirical equation accounts for the D, which stays in deuterated molecular hydrogen (HD), as it builds up to an equilibrium with HDO via the HOx-cycle.

## 4 Coupling to the hydrological cycle with the new submodel TRSYNC

In EMAC three different submodels are included dealing with isotopologues of $H_2O$ in the vapor phase: the here presented CH4 submodel, MECCA_TAG, and $H_2O$ ISOtopologues (H2OISO, Eichinger et al. (2015a)). CH4 and MECCA_TAG are treating the chemical fractionations, while H2OISO is responsible for the physical fractionations in the hydrological cycle of the underlying GCM. All three create independent tracers of $H_2O$ isotopologues, which need to be synchronized to be able to combine physical and chemical fractionation effects of $H_2O$ and its isotopologues. The chemical fractionation is thereby

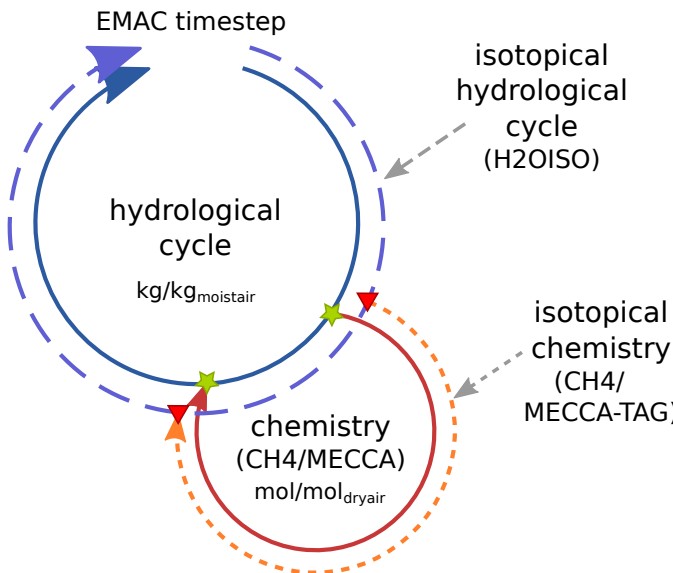

**Figure 3.** Sketch showing the coupling of the hydrological cycle and the chemistry (either CH4 or MECCA) with respect to $H_2O$ isotopologues in one time step of EMAC. Green stars indicate the points of the hydrological cycle, where (1) the current value of the water vapor master tracer is taken, and (2) the chemical tendencies are added onto the water vapor master tracer. Red triangles indicate the synchronization points of the corresponding isotopical tracers by the TRSYNC submodel. Synchronization of the isotopological cycles accounts also for the appropriate unit conversion and adds the tendency of chemical processes to the fractionation in the physical hydrological cycle.

considered either from MECCA_TAG or from CH4, although both submodels can be concurrently included in a simulation and compute the isotopic fractionation independently.

In principal, if EMAC is applied in GCM mode, only the master hydrological cycle is present (see Fig. 3, inner solid blue cycle). Adding MECCA or CH4 to the set-up expands the model into a CCM, or a simple "CH$_4$-only" CCM, respectively (red solid circle). The chemistry submodels use water vapor as a chemical tracer (first green star) and calculate the contribution from

CH$_4$ oxidation (second green star). This chemical feedback onto water vapor was already implemented as an option in previous EMAC versions. By including the isotopological submodels into the set-up, H2OISO doubles the hydrological cycle for the water isotopologues and CH4 or MECCA_TAG create the chemical tracers of the water isotopologues (outer dashed circles). This results in several physical and chemical $H_2O$ isotopologue tracers. While the master chemical process adds its feedback directly to the specific humidity of the hydrological cycle (there is no need for a chemical water tracer), the synchronization of

the physical isotopological tracers in the isotopic hydrological cycle (H2OISO) and the chemical isotopological tracers (CH4 or MECCA_TAG) is done via the new auxiliary submodel TRSYNC. In brief, TRSYNC guarantees that the physical $H_2O$ tracers (incl. their isotopologues) receive also the correct tendencies of the corresponding chemical tracers. Since isotopological water vapor tracers of MECCA_TAG and the HDO tracer created by CH4 are transported in EMAC in the same way as every other tracer, they are subject to some of the physical processes, but not to all hydrological fractionation effects. Thus, at the first





synchronization point the chemical tracer is synchronized to represent the current value of the physical tracer. In the following, chemical tendencies including fractionation effects are calculated and are added via the second synchronization point to the physical tracer. By doing so, chemical and physical fractionation processes are strictly separated and the tendencies of the chemical tracers represent the chemical tendencies in addition to the previous physical fractionations in the current time step.

Water vapor in the physical hydrological cycle (regarding ECHAM5 and H2OISO) are defined in units of kg of the tracer per

kg of moist air (kg $kg^{-1}_{moist\ air}$), while the chemical tracers are defined in mol $mol^{-1}_{dryair}$. This also holds for the corresponding isotopologue tracers. Parameterizations of physical processes in ECHAM5 are by design formulated with specific humidity (per moist air). Conversely, chemical reactions are necessarily calculated with species concentrations. This requires the individual chemical and physical isotopologue tracers, which have, for the sake of correct process formulations, distinct units, and motivated the development of the auxiliary submodel TRSYNC in order to be able to synchronize these tracers accordingly

and in a common way for CH4 and MECCA_TAG, respectively.

In addition to that, the application of MECCA_TAG creates the basis to investigate various other isotopes in the interactive chemical mechanism. While CH4 feedbacks on $H_2O$ with respect to hydrogen isotopes only, MECCA_TAG can also be used to simulate oxygen isotopes ($^{16}O$, $^{17}O$ and $^{18}O$) in the chemical mechanism. It is therefore also possible to couple MECCA_TAG with oxygen isotopes to the corresponding oxygen related isotopologue tracers in H2OISO. Last but not least,

for MECCA_TAG tracer names are not standardized. Therefore, the namelist of the submodel TRSYNC can be adjusted according to the actual tracer names used in MECCA_TAG.

## 5 Example applications

The following examples are simulations carried out with EMAC in a GCM-like mode including the newly presented CH4 and TRSYNC submodels. Other involved MESSy submodels are OFFEMIS (Kerkweg et al., 2006b) and DDEP (Kerkweg et al.,

2006a). OFFEMIS manages the emissions of $CH_4$ from prescribed sources. It reads predefined fields with emission data and adds these fluxes to the chemical tracers. DDEP simulates the dry deposition for gases and aerosols and is used in the present context to simulate the soil-loss of $CH_4$, which is not done in the CH4 submodel itself.

Monthly mean sink fields are used in the simulation set-up in the examples below. Higher frequencies are technically possible, this would, however, increase the computational demands due to the larger amount of data read from disk. Monthly mean

fields smooth the diurnal cycle, which is especially strong in OH. However, in order to investigate long-term global trends of $CH_4$, which has a tropospheric lifetime of 8–10 years, variations on time scales of less than one month are negligible and monthly mean fields are assumed to suffice for such applications. Furthermore in the examples, photolysis rates are calculated by the submodel JVAL in the presented examples, but predefined data can be used as well.

The H2OISO submodel (Eichinger, 2014; Eichinger et al., 2015a) simulates the stable water isotopologues with respect to

H and D, as well as $^{16}O$, $^{17}O$ and $^{18}O$. Overall, it represents a second hydrological cycle, which includes water isotopologues in their three phases: gas, liquid and ice. H2OISO accounts for fractionation processes during phase transitions in large scale





and convective clouds, during vertical diffusion, and during evaporation from the ocean (evaporation from soil, biosphere and snow are not considered to have a significant fractionation).

We simulated the years 1989 to 2012 and applied a specified dynamics set-up to represent the reanalyzed meteorology of this time. Specified dynamics means here that the prognostic variables divergence, vorticity, temperature and (logarithm of) surface pressure are nudged by Newtonian relaxation towards ECMWF ERA-Interim reanalysis data (Dee et al., 2011).

### 5.1   Application of the CH4 submodel for inverse optimization of $CH_4$ emission inventories

Current estimates of $CH_4$ emission inventories still include large uncertainties. In order to reduce these, new estimates of inventories must be able to represent temporal and spatial resolutions in greater detail (e.g., seasonal cycle, distinct regions). One
statistical method to estimate $CH_4$ emission strengths is the fixed-lag Kalman Filter, which performs an inverse optimization of the emission inventory by comparing simulated and observed mixing ratios of a trace gas (see e.g., Bruhwiler et al. (2005)). This "off-line" inversion algorithm requires data from a forward simulation including temporal and spatial information of the simulated $CH_4$ tracer.

In order to provide the necessary data, the CH4 submodel with the option of age and emission classes is applied. The
combination of chosen regions and emission sectors in this example results in 48 emission classes altogether. These 48 emission classes are simulated with 5 age classes for ages up to 1, 2, 3, 4, and $\geq$5 months since emission release. Figure 4 shows exemplarily the evolution of one emission class (i.e., anthropogenic emissions in Africa) from age class to age class. Panel (a) shows the emissions of the year 2000 in g($CH_4$) m$^{-2}$ per year (y$^{-1}$). The other panels (b)–(f) show the age classes in ascending order and display the distribution of the $CH_4$ mixing ratio onto the 5 age classes in January 2000 (the simulation has
started in 1989). In the fourth age class the $CH_4$ from anthropogenic African sources is almost evenly distributed mostly in the Northern Hemisphere (NH). Eventually, the fifth (i.e. the last age class) shows the accumulated background of all $CH_4$ from anthropogenic African sources. Applied is an a priori emission inventory.

Overall, the temporal evolution of the age classes in Fig. 4 confirms that the 5 age classes in this set-up sufficiently track the spread of $CH_4$ towards a fairly uniform distribution, which is a prerequisite for a successful application of the inverse
optimization method.

### 5.2   Simulating $CH_4$ isotopologues

We further present a simulation using the CH4 submodel, which includes all four $CH_4$ isotopologues. For this simulation, we applied a global a posteriori emission inventory provided by Dominik Brunner (pers. communication) and a set of isotopic emission signatures prepared from data from literature (see Table S1 in the supplement). Figure 5 shows zonal mean climatolo-
gies (2000–2009) of $CH_4$ in [nmol mol$^{-1}$] and the corresponding isotopic signature in [‰]. The isotopologues are displayed in the $\delta$-notation with respect to the reference isotope ratios Vienna-PeeDee Belemnite (VPDB) for $^{13}CH_4$, and Vienna Standard Mean Ocean Water (VSMOW) for $CH_3D$, respectively. In the troposphere the NH is isotopically depleted compared to the Southern Hemisphere (SH). Most and largest, isotopically light emissions as for example wetlands and rice are located in the NH, while isotopical heavy sources like biomass burning are mostly located in the SH. This results in the prevalent tropospheric

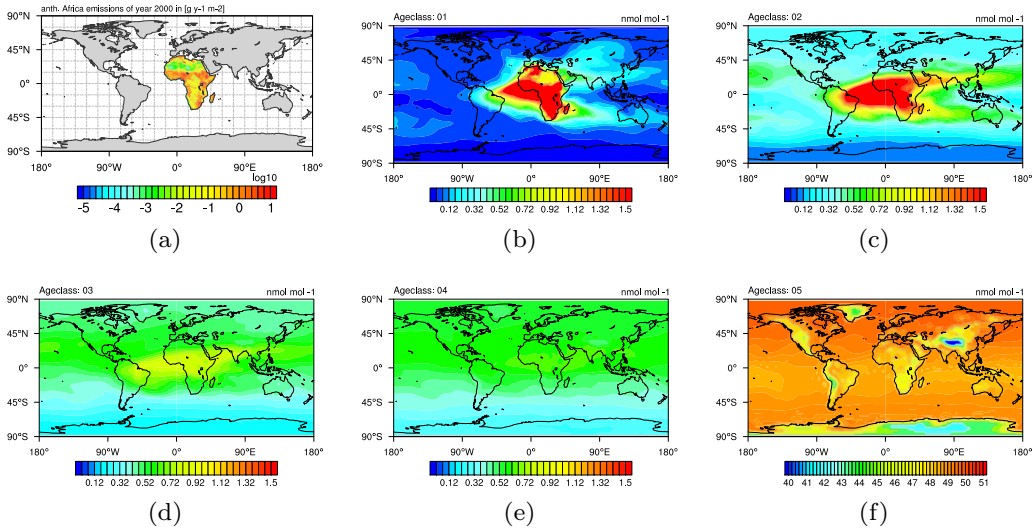

**Figure 4.** Panel (a): anthropogenic emissions in Africa (taken from EDGARv4.2 2010 fast track database (Olivier and Janssens-Maenhout, 2012)). Panels (b)–(f): Methane as pressure weighted column up to 200 hPa of anthropogenic origin from Africa, distributed into 5 age classes, i.e. up to 1, 2, 3, and 4, and ≥5 months after emission release. Shown are exemplarily all age classes of January 2000 after the simulation has run for 12 years.

North-South gradient. In the stratosphere $CH_4$ becomes isotopically enriched towards higher altitudes. This can be ascribed to fractionation processes, as heavier $CH_4$ isotopologues likely remain when $CH_4$ is exposed to oxidation during the ascend in the troposphere.

These simulation results compare well to observations. For example isotopic observations from the National Oceanic and Atmospheric Administration/Earth System Research Laboratory (NOAA/ESRL) sampling sites (White et al., 2016, 2017) and airborne samples taken during the Comprehensive Observation Network for TRace gases by AIrLiner (CONTRAIL) project (Umezawa et al., 2012) verify the North-South gradient. The values of signature of $^{13}C$ in $CH_4$ ($\delta^{13}C(CH_4)$), for example, are within the uncertainty of the CONTRAIL observations. The signature of D in $CH_4$ ($\delta D(CH_4)$) is isotopological depleted in D compared to the CONTRAIL observations, however, still capture the gradient well (not shown). The vertical gradient (i.e. isotopical enrichment in the stratosphere) can be verified by comparing with balloon borne observations by Röckmann et al. (2011). Our simulation results are thereby within the local and temporal uncertainties (not shown). Note that an optimization with respect to source signatures are yet to be made and requires an optimized emission inventory. However, the capturing of the respective gradients indicates that the isotopical fractionation is sufficiently implemented.





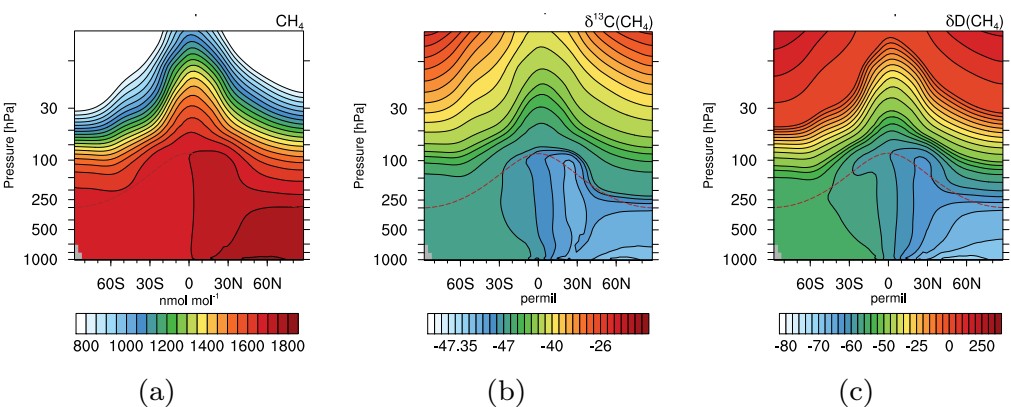

**Figure 5.** Zonal mean climatologies of 2000–2009 for $CH_4$ in [nmol mol$^{-1}$] (a), $\delta^{13}C(CH_4)$ in [‰] (b), and $\delta D(CH_4)$ in [‰] (c) of the simulation with EMAC and the CH4 submodel. The dashed brown lines indicate the height of the climatological tropopause.

### 5.3 Coupling of the $CH_4$ isotopologues to the isotopological hydrological cycle

The previously shown results were achieved with the CH4 submodel including the option to simulate $CH_4$ isotopologues. The
produced HDO (by oxidation of $CH_3D$) is connected via the TRSYNC submodel to the isotopological hydrological cycle represented by the H2OISO submodel. We carried out an additional simulation in which we applied MECCA and MECCA_TAG to simulate the atmospheric chemistry and the $CH_4$ isotopologues instead of the CH4 submodel. In this simulation TRSYNC connects the produced HDO likewise to the isotopological water tracers of H2OISO.

In Figure 6 we compare the results obtained with submodel CH4 (left) and those obtained with the submodel MECCA_TAG
(right) to vertical profiles of $H_2O$ and HDO (middle) provided by the Michelson Interferometer for Passive Atmospheric Sounding (MIPAS) instrument mounted on the ENVIronmental SATellite (ENVISAT) satellite (Steinwagner et al., 2007; Lossow et al., 2011). The ENVISAT satellite is on a sun-synchronous orbit around the Earth, completing the circuit 14 times a day. The presented observational and simulated data comprise the time period July 2002 to March 2004. The vertical range of the observations extends from 6 to 68 km (i.e. approx. the range 100–1 hPa) with a vertical resolution of 3–8 km. Simulation
and observation data is monthly and zonally averaged over the tropics. Similar to the conclusions of Eichinger et al. (2015a) it is observed that the EMAC model underestimates the $H_2O$ mixing ratio (see Figs. 6a and 6c). This is associated with a too cold tropopause in EMAC, where a temperature bias of $-2$ to $-6$ K is detected in the upper troposphere (Jöckel et al., 2016). This reduces the $H_2O$ transported into the stratosphere since more gas phase $H_2O$ freezes and sediments. Comparing Fig. 6d with 6f indicates a better agreement concerning the signature of D in $H_2O$ ($\delta D(H_2O)$) in the simulation using the submodel
MECCA_TAG with the MIPAS observations, which suggests that although the absolute $H_2O$ and HDO mixing ratios are not met, the relative composition is well represented. The differences in HDO in the simulation with the CH4 submodel compared to the one with the MECCA_TAG submodel and MIPAS are potentially caused by (1) the Eq. (9) from Eichinger et al. (2015a) used in the simulation using the CH4 submodel, which possibly does not capture important fractionation processes in the





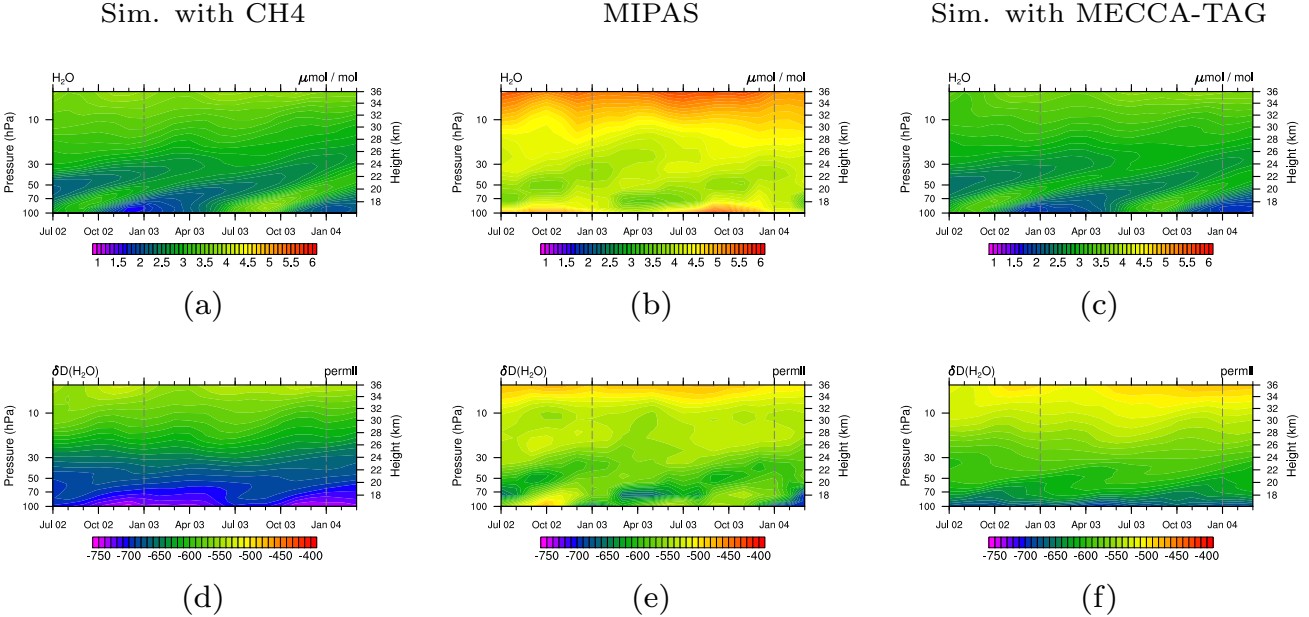

**Figure 6.** Tropical stratospheric tape recorder signal of H$_2$O (upper) and in $\delta$D(H$_2$O) (lower) in MIPAS data (middle column) and the simulations with the CH4 submodel (left column) and with the MECCA_TAG submodel (right column) in the time period July 2002 to March 2004. Simulation data is averaged monthly, zonally and over the tropics between 23° S–23° N and are displayed between 100 and 1 hPa. The grey dashed lines are included for eye guidance in the comparison of the tape recorder signal.

oxidation chain of CH$_3$D, and (2) the HD, produced in the troposphere and propagating into the stratosphere, which is not

included in the simplified chemistry, but represents an additional source of HDO. For an accurate simulation of stratospheric HDO this source needs to be considered as well in future simulations.

## 6   Summary

The submodel CH4 provides a reduced chemical set-up focusing on the CH$_4$ sink reactions, using predefined data of reaction partners, and optionally includes the feedback on SWV. This reduces the computational demands for sensitivity simulations of

climate projections without neglecting the main source of chemically induced SWV.

We presented two additional options of the CH4 submodel. The age and emission classes allow the inverse optimization of emission inventories using a fixed-lag Kalman filter. The simulation of CH$_4$ isotopologues provides further insight into the variability and distribution of CH$_4$ from its source (via emission signatures and fractionation effects) to its sink (coupling to the isotopic content of H$_2$O). The latter is implemented in form of the new submodel TRSYNC, which takes care of the correct

and time integration conform synchronization of the various H$_2$O isotopologue tracers in the model.



Example use cases show specific applications of the CH4 submodel as well as the coupling to the isotopological hydrological cycle via the TRSYNC submodel, which is especially helpful for the closure of the isotopic content in SWV.

*Code and data availability.* The Modular Earth Submodel System (MESSy) is continuously further developed and applied by a consortium of institutions. The usage of MESSy and access to the source code is licensed to all affiliates of institutions which are members of the MESSy

Consortium. Institutions can become a member of the MESSy Consortium by signing the MESSy Memorandum of Understanding. More information can be found on the MESSy Consortium Web-site (http://www.messy-interface.org). The new submodels presented in this paper have been implemented based on MESSy v2.53.0 and are available since v2.54.0. The exact code version used to produce the examples is archived at the German Climate Computing Center (DKRZ) and can be made available to members of the MESSy community upon request.

*Author contributions.* FW and PJ worked on the development of the CH4 and TRSYNC submodel and wrote the manuscript.

*Competing interests.* The authors declare that they have no conflict of interest.

*Acknowledgements.* We acknowledge the DLR internal project KliSAW (Klimarelevanz von atmosphärischen Spurengasen, Aerosolen und Wolken), which provided the financial basis for the presented model developments, and the Helmholtz-Gemeinschaft e.V. (HGF) "Project Advanced Earth System Modelling Capacity (ESM)". The model simulations have been performed at the German Climate Computing Centre (DKRZ) through support from the Bundesministerium für Bildung und Forschung (BMBF). We further thank Theresa Klausner for

her supportive comments on the manuscript.





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
