# Peer review of "Methane chemistry in a nutshell – The new submodels CH4 (v1.0) and TRSYNC (v1.0) in MESSy (v2.54.0)"

_Geoscientific Model Development, 2020_

## Referee Comment (RC1) · Anonymous Referee #1 · 10 Aug 2020

Methane chemistry in a nutshell – The new submodels CH4 (v1.0) and TRSYNC (v1.0) in MESSy (v2.54.0) , Winterstein and Joeckel, GMD-2020-137.

This GMD paper presents a description of a submodel for use in the Modular Earth Submodel System (MESSy) to describe the role of methane and its isotopologues in a reduced complexity treatment. The idea is that the CH4 submodel can be used in a variety of different configurations, some of which may choose to exclude full chemistry for reasons of computational cost, and yet use of the submodel will retain a more detailed treatment of methane and its impact than simple climatologies. The oxidant fields supplied can be from existing simulations, previously run models, or within the

simulation from more explicit chemical submodels. The paper describes the underpinning methods in the submodel, and examples where the submodels is applied to emissions inversions, simulation of isotopologues of methane and production of water isotopologues from methane oxidation.

Presentationally, the manuscript is well written and interesting. However, the model set up is mostly explained via schematics that are novel in form but regrettably not as clear as a flow chart or diagram would be. I don't feel that the diagrams, particularly Figure 1 and S1, are sufficiently clear enough to represent the mechanism or equations in use, and these should be included instead, or references supplied.

The paper describes a submodel already somewhat extensively described by Eichinger et al, 2015a, reference in this paper, and this does potentially diminish its novelty. I think it would be important to add a clear section on any differences between the implementation described here and that already in Eichinger et al.

Given that the abstract makes plain that a key area of interest is simulating stratospheric water vapor production, the lack of an assessment of the skill of the submodel in this latter regard is noticeable. I feel the paper would be strengthened, and the assessment of the submodel for SWV simulations improved, if a further section were added on this point. I appreciate that this is difficult given the underpinning model biases, but I would suggest, in particular, that the use of instantaneous production of 2 water molecules per $CH_4$ oxidised might be assessed further, and it may also be interesting to ask, What is the impact of the use of the $CH_4$ submodel on radiative forcing from all relevant species, that is $H_2O$, $CH_4$ and $O_3$ vs a model in which the effect of $CH_4$ on SWV was excluded?

Conceptually this paper has a sound idea to reduce the difficulties in simulation of methane's impact on radiative forcing to a simpler submodel, and the implementation appears to be well thought through.

I think the paper would be improved by the addition of more detail on the impact of the

choices made, particularly considering the processes or feedbacks that it was necessary to omit or treat at a reduced level of detail in the submodel and how these choices impact model skill.

Assessment of the correctness of the implementation of the atmospheric feedbacks is important here, and it is unfortunate that the concept of feedback is used somewhat broadly, which slightly obstructs the reader's own assessment of what the feedbacks are between or how they arise and whether they are implemented correctly. A key feedback is that of CH4 on OH, yet the specific examples do not mention OH, or the generation of species which could be the sink for OH, such as CO. Mention is made of HO2, however.

Similarly, the use of the phrase 'predefined fields' could be made more explicit to indicate the coupling

L7: – is the oxidation always 'offline', that is the loss of OH is not returned to the chemical solver as a feedback?

L131: the model can be coupled to, but what is the nature of the coupling? One-way (submodel receives oxidant fields) or two-way (submodel returns depleted OH, Cl fields to MECCA)?

L 138: 'secondary feedback': implies that there is feedback, but of which species? Figure 1: what do the green and black lines signify? What is the meaning of the differently shaded arrows? What is the meaning of yellow and red species?

Figure 1: caption has what I believe should be in the text 'predefined fields without feedback' – but what about the effect of HO2 on OH?

L145: Would it be possible to add what the effect of this approximation is? Does H2O feedback on stratospheric ozone?

The level of detail is mostly good: section 1 focuses on the state of our knowledge of methane chemistry and emissions, as well as the underpinning reasons and methods

for simulation of isotopologues. Section 2 descrbes the MESSy model and associate submodels. Section 3 presents the submodel in schematic form and a lengthy treatment of the treatment of the ageing of methane, which looks like new material and an important step forward from the Eichinger paper, section 4 is also new compared to Eichinger, I believe, and describes the treatment of the water isotopologues. Section 5 presents three brief examples of the application of the submodel.

Overall this is a useful description of the CH4 submodel and the improvements made since the earlier publication. I consider that t is suitable for publication having addressed these general comments and specific suggestions below.

Detailed comments

L13: what does 'similar to' mean here more precisely? What do you mean by 'feed back' to the isotopological hydrological – do you mean 'is passed back'?

L43: remove comma between both, natural

L46: what do you mean by 'not sufficiently accurate' here? Do you mean the lifetime is too short?

L56: reference required?

L60: Earth's surface

L74: rate constant not rate, given what comes after in the text

L80: k is usually reserved for rate constant but this is of course correct

L114: insert 'to' so as will read 'submodel to represent'

L186: modify 'is not conform with'

L193: modify 'to be conform with'

L200: drop comma between 'choose, whether'

L220: would make more sense as a list: 1) the CH4 submodel, 2) MECCA_TAG and 3) H2O...

L221-222: drop 'are treating'

L231: 'doubles' is not very clear: do you mean 'duplicates'?

L303: replace 'most and largest' with 'most importantly'?

L306: sentence is rather inelegant.

L308-317: values are required for quantitative comparison.

Code availability I am not an expert on Copernicus policies, but it would appear that a DOI for the code will ultimately be required.

---

## Referee Comment (RC2) · Anonymous Referee #2 · 7 Sep 2020

General comments:

This paper describes a new and simple chemical mechanism that can be run as a sub-model called CH4 within the Modular Earth Submodel System (MESSy). It represents methane and four of its most prevalent isotopologues for carbon and hydrogen by using pre-defined fields of methane sinks. It can also simulate the stratospheric water vapour (and HDO) production from methane, which can be fed back to the model's water vapour tracers. In addition, the paper describes how the CH4 sub-model can include age and emission region classes, such that the modelled trace gas concentration at a particular location in time can be attributed to particular source sectors and/or regions.

It also provides some examples of the CH4 sub-model's capability.

I thought that the manuscript was well organised and very clearly written. The schematics included in the manuscript were very informative. However, I did have one main concern (See below) which relates to the lack of an in-depth assessment of the model performance. I also include a list of more specific and technical comments - see below. If the main concern and the more detailed comments can be adequately addressed in a revised version, then the manuscript would be wholly suitable for publication in Geoscientific Model Development.

Main concern:

The CH4 model is being promoted as a useful alternative for studying methane, its isotopes, and stratospheric water vapour to the more complete and computationally expensive full chemistry scheme. As a result, I thought that the manuscript could be improved by including some verification of the CH4 model compared with the (presumably) EMAC simulation from which the sink fields used in the CH4 set up originated. How do they compare in terms of global mean methane concentration, methane lifetime, methane budget etc..? How does the modelled lifetime compare with other (full complexity) models (e.g., Stevenson et al., https://acp.copernicus.org/preprints/acp-2019-1219/) and/or inversion studies? Benchmarking the CH4 model performance against EMAC and placing its performance in the context of other models/studies would be a valuable addition to the manuscript.

The inclusion of optional region and age classes is a valuable addition to the CH4 model and this information will be useful for estimating emission strengths. The authors cite the example of using a fixed-lag Kalman Filter, which performs an inverse optimization of the emission inventory by comparing simulated and observed mixing ratios of a trace gas. However, although the example provided of the time evolution of a single region class is a nice illustration, it is by no means evidence of the suitability of the CH4 model as a tool for doing emission inventory optimization. In line with

the comment above, providing a more in-depth assessment of the model performance against observations would greatly strengthen the manuscript and provide evidence of its suitability as a tool for estimating emissions.

The authors, in the context of isotopes, also state that the simulation results compare well to observations. Can you include these comparisons with observations, for example?

Specific comments:

1. Page 2, line 25 and Page 2, line 33: While methane as a source of stratospheric water vapour (SWV) is unequivocal, it is important to, at least, acknowledge the role of methane as an ozone precursor. From a climate forcing perspective, this indirect forcing is much larger than that from methane-driven changes in SWV but is neglected from the MESSy/CH4 configuration presented here.

2. Page 8, line 190: Can you be specific about what fraction of the age class is moved to the next class when this option is used?

3. Page 8, line 194: Can you comment on how significant or large is this lack of conservation?

4. Page 14, line 331: Here, you refer to the temperature bias in EMAC leading to a negative bias in water vapour. Is this temperature bias even evident in simulations with specific dynamics or when EMAC is free running?

5. A complete listing of the CH4 chemical mechanism, including isotopes, would make the description more complete rather than only showing the temperature dependent KIEs. This could be added to the Supplementary Material.

Technical comments:

1. Page 4, line 102: Please change "The here presented new submodel for simplified CH4 chemistry (CH4) and the auxiliary submodel TRacer SYNChronization (TRSYNC)

are implemented based on this framework." to "Presented here is a new . . .."

2. Page 7, line 160: Change "which can be specified by the user via namelist" to "which can be specified by the user via a namelist"

3. Page 7, line 162: Change "denotes thereby" to "thereby denotes"

4. Page 7, line 167: Change "identical" to "identically"

5. Figure 2: The onward arrow from "tracer e02 a02" should possibly be dotted to be consistent with the one from "tracer e01 a02"

6. Page 7, line 176: Change "fixed-lag" to "a fixed time lag"

7. Page 8, line 184: The sentence "The implementation of this option is not conform with a Leapfrog time stepping with Asselin-filter and might cause numerical oscillations with negative values" Is very awkwardly written – please rephrase.

8. Page 8, line 193: Again, awkward phrasing with the use of "to be conform" in the phrase "This option is specifically implemented to be conform with the Leapfrog time stepping (c.f. option (1))". Please re-phrase.

9. Page 9, line 220: Replace "the here presented CH4 submodel" with "the CH4 submodel presented here"

10. Page 10, line 232: Replace "H2OISO doubles the hydrological cycle for the water isotopologues" with "H2OISO models the hydrological cycle for the water isotopologues" or "H2OISO represents the hydrological cycle for the water isotopologues"

---

## Author Comment (AC1) · 26 Oct 2020

**Reply to referee # 1**

October 26, 2020

Dear Referee #1,

thank you for your constructive comments on the manuscript. We appreciate your eye for detail. In the following we reply to your comments point-by-point. The indicated pages of the answers relate to the discussion paper.

**1 Specific comments**

I don't feel that the diagrams, particularly Figure 1 and S1, are sufficiently clear enough to represent the mechanism or equations in use, and these should be included instead, or references supplied.

Thank you for this suggestion. Initially, we decided to reduce the manuscript by omitting the chemical reactions included in the submodel CH4, as they are cited in nearly every publication concerning methane ($CH_4$), and provided the differential equation in form of Eq. (1) instead. However, we understand that this reduces the comprehensibility of the concept and therefore include the sink reactions of $CH_4$ in the revised manuscript and move the differential equation to the introduction section of the CH4 submodel.

The paper describes a submodel already somewhat extensively described by Eichinger et al, 2015a, reference in this paper, and this does potentially diminish its novelty. I think it would be important to add a clear section on any differences between the implementation described here and that already in Eichinger et al.

Thank you for pointing this out. It is true that Eichinger et al, 2015a used a preliminary version of the CH4 submodel. Since then the submodel was updated and extended by the age and emission classes and by the treatment of the four most abundant isotopologues (while Eichinger et al, 2015a included deuterated methane ($CH_3D$) only). In the revised manuscript, we mention these unpublished developments in the introduction. Since this manuscript in GMD is meant to be a documentation of the submodel, we think it is adequate to document all features, even if some have already been described and used by Eichinger et al. (2015), yet without a full documentation.

Included paragraph in section 1:

"An early version of the simplified $CH_4$ chemistry (CH4) submodel has been described by Eichinger et al. (2015). The present version has been updated and extended by the additional features for simulating age and emission classes and isotopologues."

I feel the paper would be strengthened, and the assessment of the submodel for SWV simulations improved, if a further section were added on this point. I appreciate that this is difficult given the underpinning model biases, but I would suggest, in particular, that the use of instantaneous production of 2 water molecules per CH4 oxidised might be assessed further, and it may also be interesting to ask, What is the impact of the use of the CH4 submodel on radiative forcing from all relevant species, that is H2O, CH4 and O3 vs a model in which the effect of CH4 on SWV was excluded?

Yes, this is an important point. We have studied the water vapor yield of $CH_4$ oxidation in detail, see Frank et al. (2018) (see also the added text in a comment below). If the CH4 submodel is used alone, there is no detailed chemical mechanism solved. Thus, in these cases there is no impact on ozone ($O_3$). Usually for such model setups a precalculated $O_3$ time series or climatology is prescribed for the radiation calculation. An evaluation of the impact of the $CH_4$ oxidation on the radiative forcing (with or without the impact on $O_3$) would be a study by itself and is clearly beyond the scope of the current manuscript, which is meant as a documentation of the submodel. Instead we refer to Revell et al. (2016), who quantified the impact of $CH_4$ oxidation on stratospheric water vapor (SWV), Stenke and Grewe (2005), who investigated the effect of SWV trends on stratospheric $O_3$ chemistry and Solomon et al. (2010), who linked changes in SWV (in particular in the upper stratosphere, where $CH_4$ oxidation makes the biggest impact) to global warming.

I think the paper would be improved by the addition of more detail on the impact of the choices made, particularly considering the processes or feedbacks that it was necessary to omit or treat at a reduced level of detail in the submodel and how these choices impact model skill.

Thank you for this suggestion. We decided to include a discussion why the present framework of a reduced chemistry is applicable to $CH_4$ and which requirements have to be met so that the simulated results are meaningful.

Included paragraph in section 3:
The presented framework of the reduced $CH_4$ chemistry is applicable, since $CH_4$ is only reduced and not produced in the free atmosphere. Therefore the discretization of the four reactions, where $CH_4$ is involved, is sufficient to represent the chemical loss of $CH_4$. Nevertheless, in order to have consistent simulation results with the CH4 submodel some prerequisites have to be met. Since the educts (the hydroxyl radical (OH), chlorine (Cl) and excited oxygen ($O(^1D)$)) are prescribed, there is no feedback on them. Thus, very large variations in $CH_4$ mixing ratio, which would in reality influence the $CH_4$ sink (Winterstein et al., 2019), are not representable by the CH4 submodel. That means it is necessary to have a balanced $CH_4$ mixing ratio and $CH_4$ sink for a sufficient simulation skill.

Assessment of the correctness of the implementation of the atmospheric feedbacks is important here, and it is unfortunate that the concept of feedback is used somewhat broadly, which slightly obstructs the reader's own assessment of what the feedbacks are between or how they arise and whether they are implemented correctly. A key feedback is that of CH4 on OH, yet the specific examples do not mention OH, or the generation of species which could be the sink for OH, such as CO. Mention is made of HO2, however.
Similarly, the use of the phrase 'predefined fields' could be made more explicit to indicate the coupling. L7: Is the oxidation always 'offline', that is the loss of OH is not returned to the chemical solver as a feedback.

Thank you for pointing this out. We see that there is need to make the phrase 'predefined fields' more clear and when we include feedbacks and when not. 'Predefined' means that they are prescribed from outside of the CH4 submodel. The CH4 submodel does not change the sink by OH (or the other sink reactants). This explains that there are no feedbacks of the CH4 submodel on the $CH_4$ sink educts and why we omitted the chemical processes forming or destroying these reactants. We added text to explain this in the manuscript (see next remark).

L131: the model can be coupled to, but what is the nature of the coupling? One-way(submodel receives oxidant fields) or two-way (submodel returns depleted OH, Cl fields to MECCA)?

> The coupling with the Module Efficiently Calculating the Chemistry of the Atmosphere (MECCA) is one way only, as the reactant fields defined by MECCA are imported into the CH4 submodel. The CH4 submodel does not alter the reactant fields (OH, Cl and O($^1$D)), but it optionally does alter the water vapor. We added this explanation in the manuscript.

The prescribed fields are taken either from existing simulation results with detailed chemistry, or from other data sources (e.g. reanalyses or projections). If CH4 is included in an ECHAM/MESSy Atmospheric Chemistry (EMAC) chemistry-climate model (CCM) simulation (which is possible in the Modular Earth Submodel System (MESSy) framework), the CH4 submodel can also be coupled to the reactant fields, which are on-line calculated during the same simulation by the chemical mechanism (i.e. MECCA). Although this does not save computational requirements, such a simulation configuration can be used, for example, if output of one of the additional options of the CH4 submodel (age and emission classes or isotopologues) are desired. In that case a second $CH_4$ tracer is treated and oxidized by the reactants solved from the kinetic solver of the comprehensive chemical mechanism. The same applies for the photolysis rate of $CH_4$, which can be prescribed from offline provided gridded data or on-line calculated by the submodel JVAL (Sander et al., 2014). **In either case, the CH4 submodel does not alter the reactant fields. Hence there is no feedback on the $CH_4$ sink by the submodel. In case of a coupling to MECCA via the reactant fields the coupling is one-way only.**

> L 138: 'secondary feedback': implies that there is feedback, but of which species?

> MECCA describes the full chemical mechanism, which includes the production and loss of the reactant species OH, Cl and O($^1$D). We rephrase this paragraph to emphasize the difference between MECCA and the CH4 submodel.

Old:
Figure 1 visualizes the conceptual differences between the MESSy submodel CH4 (left) and a CCM simulation with MECCA (right). MECCA simulates the entire chemical mechanism and therefore also includes the feedback onto the reaction partners (depicted in yellow) of $CH_4$. Additionally, there is also a secondary feedback by the products from the $CH_4$ sink reactions (e. g. water vapour ($H_2O$), $HO_2$, depicted in blue). Conversely, the CH4 submodel uses the predefined fields of the reactant species to calculate the $CH_4$ loss. This loss is included in the master tracer of the CH4 submodel, but does not feedback onto the sink fields or any other chemical species, except $H_2O$, in the case when the hydrological feedback of $CH_4$ oxidation is switched on. General Circulation Models (GCMs) include $CH_4$ foremost for its radiative impact as a greenhouse gas, but also for its influence on stratospheric water vapor (SWV, e.g. Monge-Sanz et al. (2013); ECMWF (2007); Austin et al. (2007); Boville et al. (2001); Mote (1995)). The CH4 submodel is likewise equipped with an optional feedback onto $H_2O$, to account for the secondary climate feedback of $CH_4$. It is thereby assumed that two molecules of $H_2O$ are produced per oxidized $CH_4$ molecule (le Texier et al., 1988), which is, however, only a rough approximation as analyzed by Frank et al. (2018).

New:
Figure 1 visualizes the conceptual differences between the MESSy submodel CH4 (left) and a CCM simulation with MECCA (right). MECCA simulates the entire chemical mechanism and therefore also includes the feedback onto the reaction partners (depicted in yellow) of $CH_4$. Additionally, there is also a secondary feedback by the products from the $CH_4$ sink reactions (e.g., $H_2O$, $HO_2$, depicted in blue), as the subsequent chemical processes are influenced by the products from the $CH_4$ oxidation. Conversely, the CH4 submodel uses the prescribed fields of the reactant species to calculate the $CH_4$ loss. This loss is included in the master tracer of the CH4 submodel (the present $CH_4$ is reduced), but does not feedback onto the sink fields or any other chemical species. The only exception is $H_2O$, in the case when the hydrological feedback of $CH_4$ oxidation is switched on. GCMs include $CH_4$ foremost for its radiative impact as a greenhouse gas, but also for its influence on stratospheric water vapor (SWV, e.g. Monge-Sanz et al. (2013); ECMWF (2007); Austin et al. (2007); Boville et al. (2001); Mote (1995)). The CH4 submodel is likewise equipped with an optional feedback onto $H_2O$, to account for part of the secondary climate feedback of $CH_4$. It is thereby assumed that two molecules of $H_2O$ are produced per oxidized $CH_4$ molecule (le Texier et al., 1988), which is, however, only a rough approximation as analyzed by Frank et al. (2018). The approximation of two molecules $H_2O$ per oxidized $CH_4$ molecule overestimates the

H$_2$O production in the lower stratosphere and underestimates the production in the upper stratosphere. It also does not account for the chemical loss of H$_2$O in the mesosphere.

> Figure 1: what do the green and black lines signify? What is the meaning of the differently shaded arrows? What is the meaning of yellow and red species?

> We reduced to some extent the different coloring in the figure as it has no meaning. The red species is the core species CH$_4$. We depicted the sink reactants in yellow. Blue is reserved for the products of the oxidation of CH$_4$ (H$_2$O only, in case of the CH4 submodel).

> Figure 1: caption has what I believe should be in the text 'predefined fields without feedback' but what about the effect of HO2 on OH?

> In the CH4 submodel there is no feedback of HO2 on OH. In MECCA such feedbacks are included. We changed the caption to make this more clear.

> L145: Would it be possible to add what the effect of this approximation is?

> Yes, we added a sentence describing the most important aspects of this approximation.

Included:
The constant approximation of two molecules H$_2$O per oxidized CH$_4$ molecule overestimates the H$_2$O production in the lower stratosphere and underestimates the production in the upper stratosphere. It also does not account for the chemical loss of H$_2$O in the mesosphere.

> Does H2O feedback on stratospheric ozone?

> In the case of a simulation, where the CH4 submodel is the only component simulating the atmospheric chemistry, there is no feedback of H$_2$O on O$_3$, since there is no interactively calculated O$_3$ tracer (usually only a prescribed O$_3$ climatology is used).

**2 Detailed comments**

**L13:** what does 'similar to' mean here more precisely? **We used 'similar' to point out the technical similarity in adding the produced H$_2$O and deuterated water vapour (HDO).**

What do you mean by 'feedback' to the isotopological hydrological do you mean 'is passed back'? **Thank you for this paraphrase as it is exactly what we mean. We changed it accordingly.**

**L43:** remove comma between both, natural **Agreed.**

**L46:** what do you mean by 'not sufficiently accurate' here? Do you mean the lifetime is too short? **Our intention is to state that the lifetime - or strictly speaking OH - is an important factor for the atmospheric chemistry, however challenging to simulate accurately. We rephrased this to: The lifetime of CH$_4$ is in the order of magnitude of 10 years, but its exact values is still unknown and subject to uncertainties. However, CH$_4$ is an important precursor of the Ox/HOx chemistry in CCMs. For this reason, in most CCM setups CH$_4$ is prescribed at the lower model boundary to achieve a realistic CH$_4$ burden independent of the simulated lifetime.**

**L56:** reference required? **We revised the given values and added a reference.**

**L60:** Earth's surface **Agreed.**

**L74** *and* **L80:** rate constant not rate, given what comes after in the text, k is usually reserved for rate constant but this is of course correct **Thank you for pointing this out. Although we decided to change the term to rate coefficient, since it is not constant. We removed this confusion of notation here and in the whole manuscript.**

**L114:** insert 'to' so as will read 'submodel to represent' **Agreed.**

**L186:** modify 'is not conform with' *and* **L193:** modify 'to be conform with' **We corrected *conform* by *consistent*.**

**L200:** drop comma between 'choose, whether' **Agreed.**

**L220:** would make more sense as a list: 1) the CH4 submodel, 2) MECCA_TAG and 3) H2O... **Agreed.**

**L221-222:** drop 'are treating' **We changed this to *include*.**

**L231:** 'doubles' is not very clear: do you mean 'duplicates'? **Thank you, we adopted this suggestion.**

**L303:** replace 'most and largest' with 'most importantly'? **We reduced it to "Most isotopically light emissions...", since we refer to the magnitude and extent of the emission.**

**L306:** sentence is rather inelegant. **We revised this to: "When $CH_4$ is ascending in the atmosphere it is exposed to oxidation. Due to fractionation processes heavy $CH_4$ isotopologues are unfavored and therefore accumulate in the remaining $CH_4$ content."**

**L308-317:** values are required for quantitative comparison. **We added more concrete results in the supplement.**

**References**

Austin, J., Wilson, J., Li, F., and Vömel, H.: Evolution of Water Vapor Concentrations and Stratospheric Age of Air in Coupled Chemistry-Climate Model Simulations, Am. Met. Soc., pp. 905–921, doi: 10.1175/JAS3866.1, 2007.

Boville, B. A., Kiehl, J. T., Rasch, P. J., and Bryan, F. O.: Improvements to the NCAR CSM-1 for Transient Climate Simulations, J. Climate, 14, 164–179, doi: 10.1175/1520-0442(2001)014¡0164:ITTNCF¿2.0.CO;2, 2001.

ECMWF: IFS DOCUMENTATION - Cy31r1, Part IV: Physical Processes, URL https://www.ecmwf.int/sites/default/files/elibrary/2007/9221-part-iv-physical-processes.pdf, 2007.

Eichinger, R., Jöckel, P., Brinkop, S., Werner, M., and Lossow, S.: Simulation of the isotopic composition of stratospheric water vapour - Part 1: Description and evaluation of the EMAC model, Atmos. Chem. Phys., 15, 5537–5555, doi: 10.5194/acp-15-5537-2015, 2015.

Frank, F., Jöckel, P., Gromov, S., and Dameris, M.: Investigating the yield of $H_2O$ and $H_2$ from methane oxidation in the stratosphere, Atmos. Chem. Phys., 18, 9955–9973, doi: 10.5194/acp-18-9955-2018, URL https://www.atmos-chem-phys.net/18/9955/2018/, 2018.

le Texier, H., Solomon, S., and Garcia, R. R.: The role of molecular hydrogen and methane oxidation in the water vapour budget of the stratosphere, Quart. J. Roy. Meteor. Soc., 114, 281–295, doi: 10.1002/qj.49711448002, 1988.

Monge-Sanz, B. M., Chipperfield, M. P., Untch, A., Morcrette, J.-J., Rap, A., and Simmons, A. J.: On the uses of a new linear scheme for stratospheric methane in global models: water source, transport tracer and radiative forcing, Atmos. Chem. Phys., 13, 9641–9660, doi: 10.5194/acp-13-9641-2013, URL https://www.atmos-chem-phys.net/13/9641/2013/, 2013.

Mote, P.: The annual cycle of stratospheric water vapor in a general circulation model, J. Geophys. Res., 100, 7363–7379, doi: 10.1029/94JD03301, URL http://onlinelibrary.wiley.com/doi/10.1029/94JD03301/pdf, 1995.

Revell, L., Stenke, A., Rozanov, E., Ball, W., Lossow, S., and Peter, T.: The role of methane in projections of 21st century stratospheric water vapour, Atmos. Chem. Phys., 16, 13 067–13 080, doi: 10.5194/acp-16-13067-2016, URL www.atmos-chem-phys.net/16/13067/2016/, 2016.

Sander, R., Jöckel, P., Kirner, O., Kunert, A. T., Landgraf, J., and Pozzer, A.: The photolysis module JVAL-14, compatible with the MESSy standard, and the JVal PreProcessor (JVPP), Geosci. Model Dev., 7, 2653–2662, doi: 10.5194/gmd-7-2653-2014, URL www.geosci-model-dev.net/7/2653/2014/, 2014.

Solomon, S., Rosenlof, K. H., Portmann, R. W., Daniel, J. S., Davis, S. M., Sanford, T. J., and Plattner, G.-K.: Contributions of Stratospheric Water Vapor to Decadal Changes in the Rate of Global Warming, Science, 327, 1219–1223, doi: 10.1126/science.1182488, 2010.

Stenke, A. and Grewe, V.: Simulation of stratospheric water vapor trends: impact on stratospheric ozone chemistry, Atmos. Chem. Phys., 5, 1257–1272, URL www.atmos-chem-phys.org/acp/5/1257/, 2005.

Winterstein, F., Tanalski, F., Jöckel, P., Dameris, M., and Ponater, M.: Implication of strongly increased atmospheric methane concentrations for chemistry–climate connections, Atmos. Chem. Phys., 19, 7151–7163, doi: 10.5194/acp-19-7151-2019, URL https://www.atmos-chem-phys.net/19/7151/2019/, 2019.

---

## Author Comment (AC2) · 26 Oct 2020

**Reply to referee # 2**

October 26, 2020

Dear Referee # 2,
thank you very much for such positive comments on our manuscript. In the following we reply to your comments point-by-point. The indicated pages of the answers relate to the discussion paper.

**1 Main concern**

> The CH4 model is being promoted as a useful alternative for studying methane, its isotopes, and stratospheric water vapour to the more complete and computationally expensive full chemistry scheme. As a result, I thought that the manuscript could be improved by including some verification of the CH4 model compared with the (presumably) EMAC simulation from which the sink fields used in the CH4 set up originated. How do they compare in terms of global mean methane concentration, methane lifetime, methane budget etc..? How does the modelled lifetime compare with other (fullcomplexity) models (e.g., Stevenson et al., https://acp.copernicus.org/preprints/acp-2019-1219/) and/or inversion studies? Benchmarking the CH4 model performance against EMAC and placing its performance in the context of other models/studies would be a valuable addition to the manuscript.

> Thank you for this suggestion. In fact the methane ($CH_4$) mixing ratio of the simplified $CH_4$ chemistry (CH4) submodel and the Module Efficiently Calculating the Chemistry of the Atmosphere (MECCA) are by design identical, if the same $CH_4$ sources are applied and in CH4 the same educts are prescribed as calculated in MECCA. In that case also the $CH_4$ lifetime is the same, since it is defined by the sinks. Therefore, from our point of view, a comparison of $CH_4$ simulated by the CH4 submodel with that simulated by MECCA is not really meaningful. However, an important factor for the skill of matching the atmospheric $CH_4$ mixing ratio is the method of how $CH_4$ emissions are treated. In case of prescribing $CH_4$ at the lower boundary, the $CH_4$ mixing ratio in the troposphere represents the chosen condition. In the Earth System Chemistry integrated Modelling (ESCiMo) project (Jöckel et al., 2016) the zonally averaged marine boundary surface data provided by the National Oceanic and Atmospheric Administration/Earth System Research Laboratory (NOAA/ESRL) was used as the lower boundary condition and the simulations consequently reproduced the observations. Jöckel et al. (2016) also show that the $CH_4$ lifetime in the ECHAM/MESSy Atmospheric Chemistry (EMAC) model is with $8.0 \pm 0.6$ a rather low, but within the uncertainty range of similar studies. When using emission fluxes as lower boundary condition, reproducing (globally averaged) observations is much more challenging, as current emission inventories are subject to large uncertainties and the exact lifetime of $CH_4$ is still unknown. For example, we found that inventories derived by inverse modeling are quite dependent on the assumed hydroxyl radical (OH) and hence the $CH_4$ lifetime (Frank, 2018; Zhao et al., 2020).

> The inclusion of optional region and age classes is a valuable addition to the CH4 model and this information will be useful for estimating emission strengths. The authors cite the example of using a fixed-lag Kalman Filter, which performs an inverse optimization of the emission inventory by comparing simulated and observed mixing ratios of a trace gas. However, although the example provided of the time evolution of a single region class is a nice illustration, it is by no means evidence of the suitability of the CH4 model as a tool for doing emission inventory optimization. In line with the comment above,

> providing a more in-depth assessment of the model performance against observations would greatly strengthen the manuscript and provide evidence of its suitability as a tool for estimating emissions.

> Yes, we also think that the estimation of emission strengths is a crucial part of modeling $CH_4$. The mentioned fixed-lag Kalman Filter and its application in a preproduction has been shown in Frank (2018). In the current publication we present the technical prearrangements, which are part of the CH4 submodel. As stated before, the performance of simulation results against observations is strongly influenced by the used emission inventory, which is, when targeting emission estimation, not expected to be sufficient a priori. And an in-depth analysis of the application and performance of a full inversion using the concept of the Kalman Filter would be beyond scope of the current manuscript. This will be shown elsewhere in the peer reviewed literature, since work on this is still ongoing. Nevertheless, we include the reference to Frank (2018) in the revised manuscript.

Included in section 3.1:
The third option is implemented for usage by a fixed-lag Kalman filter for inverse optimization. With this option, one age class represents one month and at the end of one month all $CH_4$ of one age class moves to the next. This option is specifically implemented to be consistent with the Leapfrog time stepping (c.f. option (1)). **A preliminary application of the concept of using the age and emission classes for an inverse optimization using the fixed-lag Kalman Filter has been shown in Frank (2018).**

> The authors, in the context of isotopes, also state that the simulation results compare well to observations. Can you include these comparisons with observations, for example?

> Yes, we added the comparisons we referred to into the revised supplement.

**2 Specific comments**

> Page 2, line 25 and Page 2, line 33: While methane as a source of stratospheric water vapour (SWV) is unequivocal, it is important to, at least, acknowledge the role of methane as an ozone precursor. From a climate forcing perspective, this indirect forcing is much larger than that from methane-driven changes in SWV but is neglected from the MESSy/CH4 configuration presented here.

> Thank you for this comment. Yes, this is indeed a drawback of the CH4 submodel and we add a discussion of this into the revised manuscript. Although we must object that the indirect forcing from influencing ozone ($O_3$) is much larger than that from water vapour ($H_2O$). From a rapid adjustments perspective the indirect forcing of $O_3$ and $H_2O$ is of about the same magnitude (Winterstein et al., 2019). Considering slow climate adjustments the effect of $H_2O$ is three times larger (Stecher et al., 2020).

Included paragraph in section 3:
Furthermore, the setup with the CH4 submodel also lacks any feedback on $O_3$. In the atmosphere, the $O_3$ chemistry is influenced by changes in the hydroxyl radical (OH) (reduced by $CH_4$), $H_2O$ (produced by $CH_4$) and temperature (influence by radiative forcing of the abundant $CH_4$). The CH4 submodel alters $H_2O$ and with that influences the radiation budget and hence the temperature, however, there is no feedback on $O_3$ when the setup does not include any other chemical mechanism. In a setup where the CH4 submodel is not used in parallel to MECCA, $O_3$ climatologies are usually prescribed for the radiation scheme.

> Page 8, line 190: Can you be specific about what fraction of the age class is moved to the next class when this option is used?

> Thank you for this question, since this seems not clear in the text. The fraction is defined by $\alpha$. We included this note to the text.

$$M' = \alpha \cdot M, \tag{1}$$

with $\alpha = \frac{\Delta t}{\tilde{T}}$ and $\tilde{T}$ being the user-defined time-span indicating the binning width of the age class. This option carries out a quasi-continuous update of the age classes, as it moves at every time step a fraction (**i.e. defined by $\alpha$**) of the current age class to the next.

> Page 8, line 194: Can you comment on how significant or large is this lack of conservation?

> The described procedure is done to avoid the accumulation of small (numerical) errors, which mainly arise from small non-linearities of the large scale advection scheme. The magnitude therefore depends on the applied advection scheme, but is usually of the order of floating point precision. We added this explanation to the text as well.

Included in section 3.1:
In order to reduce numerical errors, the age and emission classes are continuously constrained (i.e., in each model time step) to sum up to the master tracer and are scaled appropriately, if the sum deviates. **The described procedure is done to avoid the accumulation of such numerical errors, which mainly arise from small non-linearities of the large scale advection scheme. The magnitude therefore depends on the applied advection scheme, but is usually of the order of floating point precision.**

> Page 14, line 331: Here, you refer to the temperature bias in EMAC leading to a negative bias in water vapour. Is this temperature bias even evident in simulations with specific dynamics or when EMAC is free running?

> The negative temperature bias in EMAC is strongest in free running set-ups. It is reduced but is still evident in simulations with specified dynamics as long as the wave-0 (or mean) of the temperature is not included in the nudging procedure, i.e. the temperature bias is not corrected. This is the usually applied procedure for specified dynamics. As soon as the mean temperature is included in the nudging, the bias nearly disappears. For more detailed information on the nudging procedure and the temperature bias, we refer to Jöckel et al. (2016).

Included in section 5.3:
This is associated with a too cold tropopause in EMAC, where a temperature bias of $-2$ to $-6$ K is detected in the upper troposphere, **as long as the mean temperature is excluded from the nudging procedure defining the specified dynamics setup** (Jöckel et al., 2016).

> A complete listing of the CH4 chemical mechanism, including isotopes, would make the description more complete rather than only showing the temperature dependent KIEs. This could be added to the Supplementary Material.

> As also suggested by the other reviewer we include in the revision the $CH_4$ sink reactions (R1–R4) in section 1. We also include the corresponding reactions with isotopes deuterium (D) and carbon-13 ($^{13}$C) in the revised supplement.

**3   Technical comments**

**Page 4, line 102:** Please change The here presented new submodel for simplified CH4 chemistry (CH4) and the auxiliary submodel TRacer SYNChronization (TRSYNC) are implemented based on this framework. to Presented here is a new.... **Agreed.**

**Page 7, line 160:** Change which can be specified by the user via namelist to which can be specified by the user via a namelist **Agreed.**

**Page 7, line 162:** Change denotes thereby to thereby denotes **Agreed.**

**Page 7, line 167:** Change identical to identically **Agreed.**

**Figure 2:** The onward arrow from tracer e02 a02 should possibly be dotted to be consistent with the one from tracer e01 a02 **Thank you, we changed that for consistency.**

**Page 7, line 176:** Change fixed-lag to a fixed time lag **Agreed.**

**Page 8, line 184:** The sentence The implementation of this option is not conform with a Leapfrog time stepping with Asselin-filter and might cause numerical oscillations with negative values Is very awkwardly written  please rephrase. **We changed it to: This option is not consistent with a Leapfrog time stepping using an Asselin-filter and might cause numerical oscillations and negative values.**

**Page 8, line 193:** Again, awkward phrasing with the use of to be conform in the phrase This option is specifically implemented to be conform with the Leapfrog timestepping (c.f. option (1)). Please re-phrase. **We corrected *conform* by *consistent.***

**Page 9, line 220:** Replace the here presented CH4 submodel with the CH4 submodel presented here **Agreed.**

**Page 10, line 232:** Replace H2OISO doubles the hydrological cycle for the water isotopologues with H2OISO models the hydrological cycle for the water isotopologues or H2OISO represents the hydrological cycle for the water isotopologues **We changed *doubles* to *dublicates*. We want to point out that the hydrological cycle in H2OISO is in addition to the cycle in ECHAM.**

Thank you for these suggestions and corrections. We changed the manuscript accordingly.

**References**

Frank, F.: Atmospheric methane and its isotopic composition in a changing climate: A modelling study, Ph.D. thesis, Ludwigs Maximillian Universität München, 2018.

Jöckel, P., Tost, H., Pozzer, A., Kunze, M., Kirner, O., Brenninkmeijer, C. A. M., Brinkop, S., Cai, D. S., Dyroff, C., Eckstein, J., Frank, F., Garny, H., Gottschaldt, K.-D., Graf, P., Grewe, V., , Kerkweg, A., Kern, B., Matthes, S., Mertens, M., Meul, S., Neumaier, M., Nützel, M., Oberländer-Hayn, S., Ruhnke, R., Runde, T., Sander, R., Scharffe, D., and Zahn, A.: Earth System Chemistry integrated Modelling (ESCiMo) with the Modular Earth Submodel System (MESSy) version 2.51, Geosci. Model Dev., 9, 1153–1200, doi: 10.5194/gmd-9-1153-2016, URL http://www.geosci-model-dev.net/9/1153/2016/gmd-9-1153-2016.html, 2016.

Stecher, L., Winterstein, F., Dameris, M., Jöckel, P., Ponater, M., and Kunze, M.: Effects of Strongly Enhanced Atmospheric Methane Concentrations in a Fully Coupled Chemistry-Climate Model, Atmospheric Chemistry and Physics Discussions, 2020, 1–31, doi: 10.5194/acp-2020-519, URL https://acp.copernicus.org/preprints/acp-2020-519/, 2020.

Winterstein, F., Tanalski, F., Jöckel, P., Dameris, M., and Ponater, M.: Implication of strongly increased atmospheric methane concentrations for chemistry–climate connections, Atmos. Chem. Phys., 19, 7151–7163, doi: 10.5194/acp-19-7151-2019, URL `https://www.atmos-chem-phys.net/19/7151/2019/`, 2019.

Zhao, Y., Saunois, M., Bousquet, P., Lin, X., Berchet, A., Hegglin, M. I., Canadell, J. G., Jackson, R. B., Deushi, M., Jöckel, P., Kinnison, D., Kirner, O., Strode, S., Tilmes, S., Dlugokencky, E. J., and Zheng, B.: On the role of trend and variability of hydroxyl radical (OH) in the global methane budget, Atmospheric Chemistry and Physics Discussions, 2020, 1–28, doi: 10.5194/acp-2020-308, URL `https://acp.copernicus.org/preprints/acp-2020-308/`, 2020.

---

## Author Response (AR1)

**Author's response**

November 5, 2020

Dear Fiona O'Connor,

thank you for overseeing the review process of our manuscript. In the following we attached our reply to the reviewer comments. The revised manuscript and supplement with all changes highlighted is included at the end of this document.

If any questions arise, please do not hesitate to contact me.

Kind regards,
Franziska Winterstein, on behalf of all co-authors

**Reply to referee # 1**

October 26, 2020

Dear Referee #1,

thank you for your constructive comments on the manuscript. We appreciate your eye for detail. In the following we reply to your comments point-by-point. The indicated pages of the answers relate to the discussion paper.

**1 Specific comments**

> I don't feel that the diagrams, particularly Figure 1 and S1, are sufficiently clear enough to represent the mechanism or equations in use, and these should be included instead, or references supplied.

> Thank you for this suggestion. Initially, we decided to reduce the manuscript by omitting the chemical reactions included in the submodel CH4, as they are cited in nearly every publication concerning methane ($CH_4$), and provided the differential equation in form of Eq. (1) instead. However, we understand that this reduces the comprehensibility of the concept and therefore include the sink reactions of $CH_4$ in the revised manuscript and move the differential equation to the introduction section of the CH4 submodel.

> The paper describes a submodel already somewhat extensively described by Eichinger et al, 2015a, reference in this paper, and this does potentially diminish its novelty. I think it would be important to add a clear section on any differences between the implementation described here and that already in Eichinger et al.

> Thank you for pointing this out. It is true that Eichinger et al, 2015a used a preliminary version of the CH4 submodel. Since then the submodel was updated and extended by the age and emission classes and by the treatment of the four most abundant isotopologues (while Eichinger et al, 2015a included deuterated methane ($CH_3D$) only). In the revised manuscript, we mention these unpublished developments in the introduction. Since this manuscript in GMD is meant to be a documentation of the submodel, we think it is adequate to document all features, even if some have already been described and used by Eichinger et al. (2015), yet without a full documentation.

Included paragraph in section 1:

"An early version of the simplified $CH_4$ chemistry (CH4) submodel has been described by Eichinger et al. (2015). The present version has been updated and extended by the additional features for simulating age and emission classes and isotopologues."

> I feel the paper would be strengthened, and the assessment of the submodel for SWV simulations improved, if a further section were added on this point. I appreciate that this is difficult given the underpinning model biases, but I would suggest, in particular, that the use of instantaneous production of 2 water molecules per CH4 oxidised might be assessed further, and it may also be interesting to ask, What is the impact of the use of the CH4 submodel on radiative forcing from all relevant species, that is H2O, CH4 and O3 vs a model in which the effect of CH4 on SWV was excluded?

Yes, this is an important point. We have studied the water vapor yield of $CH_4$ oxidation in detail, see Frank et al. (2018) (see also the added text in a comment below). If the CH4 submodel is used alone, there is no detailed chemical mechanism solved. Thus, in these cases there is no impact on ozone ($O_3$). Usually for such model setups a precalculated $O_3$ time series or climatology is prescribed for the radiation calculation. An evaluation of the impact of the $CH_4$ oxidation on the radiative forcing (with or without the impact on $O_3$) would be a study by itself and is clearly beyond the scope of the current manuscript, which is meant as a documentation of the submodel. Instead we refer to Revell et al. (2016), who quantified the impact of $CH_4$ oxidation on stratospheric water vapor (SWV), Stenke and Grewe (2005), who investigated the effect of SWV trends on stratospheric $O_3$ chemistry and Solomon et al. (2010), who linked changes in SWV (in particular in the upper stratosphere, where $CH_4$ oxidation makes the biggest impact) to global warming.

I think the paper would be improved by the addition of more detail on the impact of the choices made, particularly considering the processes or feedbacks that it was necessary to omit or treat at a reduced level of detail in the submodel and how these choices impact model skill.

Thank you for this suggestion. We decided to include a discussion why the present framework of a reduced chemistry is applicable to $CH_4$ and which requirements have to be met so that the simulated results are meaningful.

Included paragraph in section 3:
The presented framework of the reduced $CH_4$ chemistry is applicable, since $CH_4$ is only reduced and not produced in the free atmosphere. Therefore the discretization of the four reactions, where $CH_4$ is involved, is sufficient to represent the chemical loss of $CH_4$. Nevertheless, in order to have consistent simulation results with the CH4 submodel some prerequisites have to be met. Since the educts (the hydroxyl radical (OH), chlorine (Cl) and excited oxygen ($O(^1D)$)) are prescribed, there is no feedback on them. Thus, very large variations in $CH_4$ mixing ratio, which would in reality influence the $CH_4$ sink (Winterstein et al., 2019), are not representable by the CH4 submodel. That means it is necessary to have a balanced $CH_4$ mixing ratio and $CH_4$ sink for a sufficient simulation skill.

Assessment of the correctness of the implementation of the atmospheric feedbacks is important here, and it is unfortunate that the concept of feedback is used somewhat broadly, which slightly obstructs the reader's own assessment of what the feedbacks are between or how they arise and whether they are implemented correctly. A key feedback is that of CH4 on OH, yet the specific examples do not mention OH, or the generation of species which could be the sink for OH, such as CO. Mention is made of HO2, however.
Similarly, the use of the phrase 'predefined fields' could be made more explicit to indicate the coupling.
L7: Is the oxidation always 'offline', that is the loss of OH is not returned to the chemical solver as a feedback.

Thank you for pointing this out. We see that there is need to make the phrase 'predefined fields' more clear and when we include feedbacks and when not. 'Predefined' means that they are prescribed from outside of the CH4 submodel. The CH4 submodel does not change the sink by OH (or the other sink reactants). This explains that there are no feedbacks of the CH4 submodel on the $CH_4$ sink educts and why we omitted the chemical processes forming or destroying these reactants. We added text to explain this in the manuscript (see next remark).

L131: the model can be coupled to, but what is the nature of the coupling? One-way(submodel receives oxidant fields) or two-way (submodel returns depleted OH, Cl fields to MECCA)?

> The coupling with the Module Efficiently Calculating the Chemistry of the Atmosphere (MECCA) is one way only, as the reactant fields defined by MECCA are imported into the CH4 submodel. The CH4 submodel does not alter the reactant fields (OH, Cl and O($^1$D)), but it optionally does alter the water vapor. We added this explanation in the manuscript.

The prescribed fields are taken either from existing simulation results with detailed chemistry, or from other data sources (e.g. reanalyses or projections). If CH4 is included in an ECHAM/MESSy Atmospheric Chemistry (EMAC) chemistry-climate model (CCM) simulation (which is possible in the Modular Earth Submodel System (MESSy) framework), the CH4 submodel can also be coupled to the reactant fields, which are on-line calculated during the same simulation by the chemical mechanism (i.e. MECCA). Although this does not save computational requirements, such a simulation configuration can be used, for example, if output of one of the additional options of the CH4 submodel (age and emission classes or isotopologues) are desired. In that case a second $CH_4$ tracer is treated and oxidized by the reactants solved from the kinetic solver of the comprehensive chemical mechanism. The same applies for the photolysis rate of $CH_4$, which can be prescribed from offline provided gridded data or on-line calculated by the submodel JVAL (Sander et al., 2014). **In either case, the CH4 submodel does not alter the reactant fields. Hence there is no feedback on the $CH_4$ sink by the submodel. In case of a coupling to MECCA via the reactant fields the coupling is one-way only.**

> L 138: 'secondary feedback': implies that there is feedback, but of which species?

> MECCA describes the full chemical mechanism, which includes the production and loss of the reactant species OH, Cl and O($^1$D). We rephrase this paragraph to emphasize the difference between MECCA and the CH4 submodel.

Old:
Figure 1 visualizes the conceptual differences between the MESSy submodel CH4 (left) and a CCM simulation with MECCA (right). MECCA simulates the entire chemical mechanism and therefore also includes the feedback onto the reaction partners (depicted in yellow) of $CH_4$. Additionally, there is also a secondary feedback by the products from the $CH_4$ sink reactions (e. g. water vapour ($H_2O$), $HO_2$, depicted in blue). Conversely, the CH4 submodel uses the predefined fields of the reactant species to calculate the $CH_4$ loss. This loss is included in the master tracer of the CH4 submodel, but does not feedback onto the sink fields or any other chemical species, except $H_2O$, in the case when the hydrological feedback of $CH_4$ oxidation is switched on. General Circulation Models (GCMs) include $CH_4$ foremost for its radiative impact as a greenhouse gas, but also for its influence on stratospheric water vapor (SWV, e.g. Monge-Sanz et al. (2013); ECMWF (2007); Austin et al. (2007); Boville et al. (2001); Mote (1995)). The CH4 submodel is likewise equipped with an optional feedback onto $H_2O$, to account for the secondary climate feedback of $CH_4$. It is thereby assumed that two molecules of $H_2O$ are produced per oxidized $CH_4$ molecule (le Texier et al., 1988), which is, however, only a rough approximation as analyzed by Frank et al. (2018).

New:
Figure 1 visualizes the conceptual differences between the MESSy submodel CH4 (left) and a CCM simulation with MECCA (right). MECCA simulates the entire chemical mechanism and therefore also includes the feedback onto the reaction partners (depicted in yellow) of $CH_4$. Additionally, there is also a secondary feedback by the products from the $CH_4$ sink reactions (e.g., $H_2O$, $HO_2$, depicted in blue), as the subsequent chemical processes are influenced by the products from the $CH_4$ oxidation. Conversely, the CH4 submodel uses the prescribed fields of the reactant species to calculate the $CH_4$ loss. This loss is included in the master tracer of the CH4 submodel (the present $CH_4$ is reduced), but does not feedback onto the sink fields or any other chemical species. The only exception is $H_2O$, in the case when the hydrological feedback of $CH_4$ oxidation is switched on. GCMs include $CH_4$ foremost for its radiative impact as a greenhouse gas, but also for its influence on stratospheric water vapor (SWV, e.g. Monge-Sanz et al. (2013); ECMWF (2007); Austin et al. (2007); Boville et al. (2001); Mote (1995)). The CH4 submodel is likewise equipped with an optional feedback onto $H_2O$, to account for part of the secondary climate feedback of $CH_4$. It is thereby assumed that two molecules of $H_2O$ are produced per oxidized $CH_4$ molecule (le Texier et al., 1988), which is, however, only a rough approximation as analyzed by Frank et al. (2018). The approximation of two molecules $H_2O$ per oxidized $CH_4$ molecule overestimates the

$H_2O$ production in the lower stratosphere and underestimates the production in the upper stratosphere. It also does not account for the chemical loss of $H_2O$ in the mesosphere.

> Figure 1: what do the green and black lines signify? What is the meaning of the differently shaded arrows? What is the meaning of yellow and red species?

> We reduced to some extent the different coloring in the figure as it has no meaning. The red species is the core species $CH_4$. We depicted the sink reactants in yellow. Blue is reserved for the products of the oxidation of $CH_4$ ($H_2O$ only, in case of the CH4 submodel).

> Figure 1: caption has what I believe should be in the text 'predefined fields without feedback' but what about the effect of HO2 on OH?

> In the CH4 submodel there is no feedback of HO2 on OH. In MECCA such feedbacks are included. We changed the caption to make this more clear.

> L145: Would it be possible to add what the effect of this approximation is?

> Yes, we added a sentence describing the most important aspects of this approximation.

Included:
The constant approximation of two molecules $H_2O$ per oxidized $CH_4$ molecule overestimates the $H_2O$ production in the lower stratosphere and underestimates the production in the upper stratosphere. It also does not account for the chemical loss of $H_2O$ in the mesosphere.

> Does H2O feedback on stratospheric ozone?

> In the case of a simulation, where the CH4 submodel is the only component simulating the atmospheric chemistry, there is no feedback of $H_2O$ on $O_3$, since there is no interactively calculated $O_3$ tracer (usually only a prescribed $O_3$ climatology is used).

**2    Detailed comments**

**L13:** what does 'similar to' mean here more precisely? **We used 'similar' to point out the technical similarity in adding the produced $H_2O$ and deuterated water vapour (HDO).**

What do you mean by 'feedback' to the isotopological hydrological  do you mean 'is passed back'? **Thank you for this paraphrase as it is exactly what we mean. We changed it accordingly.**

**L43:** remove comma between both, natural **Agreed.**

**L46:** what do you mean by 'not sufficiently accurate' here? Do you mean the lifetime is too short? **Our intention is to state that the lifetime - or strictly speaking OH - is an important factor for the atmospheric chemistry, however challenging to simulate accurately. We rephrased this to: The lifetime of $CH_4$ is in the order of magnitude of 10 years, but its exact values is still unknown and subject to uncertainties. However, $CH_4$ is an important precursor of the Ox/HOx chemistry in CCMs. For this reason, in most CCM setups $CH_4$ is prescribed at the lower model boundary to achieve a realistic $CH_4$ burden independent of the simulated lifetime.**

**L56:** reference required? **We revised the given values and added a reference.**

**L60:** Earth's surface **Agreed.**

**L74** *and* **L80:** rate constant not rate, given what comes after in the text, k is usually reserved for rate constant but this is of course correct **Thank you for pointing this out. Although we decided to change the term to rate coefficient, since it is not constant. We removed this confusion of notation here and in the whole manuscript.**

**L114:** insert 'to' so as will read 'submodel to represent' **Agreed.**

**L186:** modify 'is not conform with' *and* **L193:** modify 'to be conform with' **We corrected *conform* by *consistent.***

**L200:** drop comma between 'choose, whether' **Agreed.**

**L220:** would make more sense as a list: 1) the CH4 submodel, 2) MECCA_TAG and 3) H2O... **Agreed.**

**L221-222:** drop 'are treating' **We changed this to *include.***

**L231:** 'doubles' is not very clear: do you mean 'duplicates'? **Thank you, we adopted this suggestion.**

**L303:** replace 'most and largest' with 'most importantly'? **We reduced it to "Most isotopically light emissions...", since we refer to the magnitude and extent of the emission.**

**L306:** sentence is rather inelegant. **We revised this to: "When $CH_4$ is ascending in the atmosphere it is exposed to oxidation. Due to fractionation processes heavy $CH_4$ isotopologues are unfavored and therefore accumulate in the remaining $CH_4$ content."**

**L308-317:** values are required for quantitative comparison. **We added more concrete results in the supplement.**

Thank you for this suggestion. In fact the methane ($CH_4$) mixing ratio of the simplified $CH_4$ chemistry (CH4) submodel and the Module Efficiently Calculating the Chemistry of the Atmosphere (MECCA) are by design identical, if the same $CH_4$ sources are applied and in CH4 the same educts are prescribed as calculated in MECCA. In that case also the $CH_4$ lifetime is the same, since it is defined by the sinks. Therefore, from our point of view, a comparison of $CH_4$ simulated by the CH4 submodel with that simulated by MECCA is not really meaningful. However, an important factor for the skill of matching the atmospheric $CH_4$ mixing ratio is the method of how $CH_4$ emissions are treated. In case of prescribing $CH_4$ at the lower boundary, the $CH_4$ mixing ratio in the troposphere represents the chosen condition. In the Earth System Chemistry integrated Modelling (ESCiMo) project (Jöckel et al., 2016) the zonally averaged marine boundary surface data provided by the National Oceanic and Atmospheric Administration/Earth System Research Laboratory (NOAA/ESRL) was used as the lower boundary condition and the simulations consequently reproduced the observations. Jöckel et al. (2016) also show that the $CH_4$ lifetime in the ECHAM/MESSy Atmospheric Chemistry (EMAC) model is with $8.0 \pm 0.6$ a rather low, but within the uncertainty range of similar studies. When using emission fluxes as lower boundary condition, reproducing (globally averaged) observations is much more challenging, as current emission inventories are subject to large uncertainties and the exact lifetime of $CH_4$ is still unknown. For example, we found that inventories derived by inverse modeling are quite dependent on the assumed hydroxyl radical (OH) and hence the $CH_4$ lifetime (Frank, 2018; Zhao et al., 2020).

The inclusion of optional region and age classes is a valuable addition to the CH4 model and this information will be useful for estimating emission strengths. The authors cite the example of using a fixed-lag Kalman Filter, which performs an inverse optimization of the emission inventory by comparing simulated and observed mixing ratios of a trace gas. However, although the example provided of the time evolution of a single region class is a nice illustration, it is by no means evidence of the suitability of the CH4 model as a tool for doing emission inventory optimization. In line with the comment above,

> providing a more in-depth assessment of the model performance against observations would greatly strengthen the manuscript and provide evidence of its suitability as a tool for estimating emissions.

> Yes, we also think that the estimation of emission strengths is a crucial part of modeling $CH_4$. The mentioned fixed-lag Kalman Filter and its application in a preproduction has been shown in Frank (2018). In the current publication we present the technical prearrangements, which are part of the CH4 submodel. As stated before, the performance of simulation results against observations is strongly influenced by the used emission inventory, which is, when targeting emission estimation, not expected to be sufficient a priori. And an in-depth analysis of the application and performance of a full inversion using the concept of the Kalman Filter would be beyond scope of the current manuscript. This will be shown elsewhere in the peer reviewed literature, since work on this is still ongoing. Nevertheless, we include the reference to Frank (2018) in the revised manuscript.

Included in section 3.1:
The third option is implemented for usage by a fixed-lag Kalman filter for inverse optimization. With this option, one age class represents one month and at the end of one month all $CH_4$ of one age class moves to the next. This option is specifically implemented to be consistent with the Leapfrog time stepping (c.f. option (1)). **A preliminary application of the concept of using the age and emission classes for an inverse optimization using the fixed-lag Kalman Filter has been shown in Frank (2018).**

> The authors, in the context of isotopes, also state that the simulation results compare well to observations. Can you include these comparisons with observations, for example?

> Yes, we added the comparisons we referred to into the revised supplement.

**2 Specific comments**

> Page 2, line 25 and Page 2, line 33: While methane as a source of stratospheric water vapour (SWV) is unequivocal, it is important to, at least, acknowledge the role of methane as an ozone precursor. From a climate forcing perspective, this indirect forcing is much larger than that from methane-driven changes in SWV but is neglected from the MESSy/CH4 configuration presented here.

> Thank you for this comment. Yes, this is indeed a drawback of the CH4 submodel and we add a discussion of this into the revised manuscript. Although we must object that the indirect forcing from influencing ozone ($O_3$) is much larger than that from water vapour ($H_2O$). From a rapid adjustments perspective the indirect forcing of $O_3$ and $H_2O$ is of about the same magnitude (Winterstein et al., 2019). Considering slow climate adjustments the effect of $H_2O$ is three times larger (Stecher et al., 2020).

Included paragraph in section 3:
Furthermore, the setup with the CH4 submodel also lacks any feedback on $O_3$. In the atmosphere, the $O_3$ chemistry is influenced by changes in the hydroxyl radical (OH) (reduced by $CH_4$), $H_2O$ (produced by $CH_4$) and temperature (influence by radiative forcing of the abundant $CH_4$). The CH4 submodel alters $H_2O$ and with that influences the radiation budget and hence the temperature, however, there is no feedback on $O_3$ when the setup does not include any other chemical mechanism. In a setup where the CH4 submodel is not used in parallel to MECCA, $O_3$ climatologies are usually prescribed for the radiation scheme.

> Page 8, line 190: Can you be specific about what fraction of the age class is moved to the next class when this option is used?

Thank you for this question, since this seems not clear in the text. The fraction is defined by $\alpha$. We included this note to the text.

$$M' = \alpha \cdot M, \tag{1}$$

with $\alpha = \frac{\Delta t}{\tilde{T}}$ and $\tilde{T}$ being the user-defined time-span indicating the binning width of the age class. This option carries out a quasi-continuous update of the age classes, as it moves at every time step a fraction (**i.e. defined by $\alpha$**) of the current age class to the next.

Page 8, line 194: Can you comment on how significant or large is this lack of conservation?

The described procedure is done to avoid the accumulation of small (numerical) errors, which mainly arise from small non-linearities of the large scale advection scheme. The magnitude therefore depends on the applied advection scheme, but is usually of the order of floating point precision. We added this explanation to the text as well.

Included in section 3.1:
In order to reduce numerical errors, the age and emission classes are continuously constrained (i.e., in each model time step) to sum up to the master tracer and are scaled appropriately, if the sum deviates. **The described procedure is done to avoid the accumulation of such numerical errors, which mainly arise from small non-linearities of the large scale advection scheme. The magnitude therefore depends on the applied advection scheme, but is usually of the order of floating point precision.**

Page 14, line 331: Here, you refer to the temperature bias in EMAC leading to a negative bias in water vapour. Is this temperature bias even evident in simulations with specific dynamics or when EMAC is free running?

The negative temperature bias in EMAC is strongest in free running set-ups. It is reduced but is still evident in simulations with specified dynamics as long as the wave-0 (or mean) of the temperature is not included in the nudging procedure, i.e. the temperature bias is not corrected. This is the usually applied procedure for specified dynamics. As soon as the mean temperature is included in the nudging, the bias nearly disappears. For more detailed information on the nudging procedure and the temperature bias, we refer to Jöckel et al. (2016).

Included in section 5.3:
This is associated with a too cold tropopause in EMAC, where a temperature bias of $-2$ to $-6$ K is detected in the upper troposphere, **as long as the mean temperature is excluded from the nudging procedure defining the specified dynamics setup** (Jöckel et al., 2016).

A complete listing of the CH4 chemical mechanism, including isotopes, would make the description more complete rather than only showing the temperature dependent KIEs. This could be added to the Supplementary Material.

As also suggested by the other reviewer we include in the revision the $CH_4$ sink reactions (R1–R4) in section 1. We also include the corresponding reactions with isotopes deuterium (D) and carbon-13 ($^{13}$C) in the revised supplement.

**3 Technical comments**

**Page 4, line 102:** Please change The here presented new submodel for simplified CH4 chemistry (CH4) and the auxiliary submodel TRacer SYNChronization (TRSYNC) are implemented based on this framework. to Presented here is a new.... **Agreed.**

**Page 7, line 160:** Change which can be specified by the user via namelist to which can be specified by the user via a namelist **Agreed.**

**Page 7, line 162:** Change denotes thereby to thereby denotes **Agreed.**

**Page 7, line 167:** Change identical to identically **Agreed.**

**Figure 2:** The onward arrow from tracer e02 a02 should possibly be dotted to be consistent with the one from tracer e01 a02 **Thank you, we changed that for consistency.**

**Page 7, line 176:** Change fixed-lag to a fixed time lag **Agreed.**

**Page 8, line 184:** The sentence The implementation of this option is not conform with a Leapfrog time stepping with Asselin-filter and might cause numerical oscillations with negative values Is very awkwardly written please rephrase. **We changed it to: This option is not consistent with a Leapfrog time stepping using an Asselin-filter and might cause numerical oscillations and negative values.**

**Page 8, line 193:** Again, awkward phrasing with the use of to be conform in the phrase This option is specifically implemented to be conform with the Leapfrog timestepping (c.f. option (1)). Please re-phrase. **We corrected *conform* by *consistent.***

**Page 9, line 220:** Replace the here presented CH4 submodel with the CH4 submodel presented here **Agreed.**

**Page 10, line 232:** Replace H2OISO doubles the hydrological cycle for the water isotopologues with H2OISO models the hydrological cycle for the water isotopologues or H2OISO represents the hydrological cycle for the water isotopologues **We changed *doubles* to *dublicates*. We want to point out that the hydrological cycle in H2OISO is in addition to the cycle in ECHAM.**

Thank you for these suggestions and corrections. We changed the manuscript accordingly.

[revised manuscript text omitted]

**1 Chemical processes and reaction rate coefficients concerning $CH_4$**

35 ## 1.1 Sink reactions

General sink reactions:

$$CH_4 + OH \quad \overset{k_{CH_4+OH}}{\rightarrow} \quad CH_3 + H_2O \tag{SR1}$$

$$CH_4 + O(^1D) \quad \overset{k_{CH_4+O(^1D)}}{\rightarrow} \quad products \tag{SR2}$$

$$CH_4 + Cl \quad \overset{k_{CH_4+Cl}}{\rightarrow} \quad CH_3 + HCl \tag{SR3}$$

$$CH_4 + h\nu \quad \overset{k_{CH_4+h\nu}}{\rightarrow} \quad products \tag{SR4}$$

Sink reactions with isotopologues containing carbon-13 ($^{13}$C) :

$$^{13}CH_4 + OH \quad \overset{k_{^{13}CH_4+OH}}{\rightarrow} \quad {}^{13}CH_3 + H_2O \tag{SR5}$$

45 $$^{13}CH_4 + O(^1D) \quad \overset{k_{^{13}CH_4+O(^1D)}}{\rightarrow} \quad products \tag{SR6}$$

$$^{13}CH_4 + Cl \quad \overset{k_{^{13}CH_4+Cl}}{\rightarrow} \quad {}^{13}CH_3 + HCl \tag{SR7}$$

$$^{13}CH_4 + h\nu \quad \overset{k_{^{13}CH_4+h\nu}}{\rightarrow} \quad products \tag{SR8}$$

Sink reactions with isotopologues containing deuterium (D):

$$CH_3D + OH \quad \overset{k_{CH_3D+OH}}{\rightarrow} \quad CH_2D + H_2O \tag{SR9}$$

$$CH_3 + HDO \tag{SR10}$$

$$CH_3D + O(^1D) \quad \overset{k_{CH_3D+O(^1D)}}{\rightarrow} \quad products \tag{SR11}$$

$$CH_3D + Cl \quad \overset{k_{CH_3D+Cl}}{\rightarrow} \quad CH_2D + HCl \tag{SR12}$$

$$CH_3 + DCl \tag{SR13}$$

$$CH_3D + h\nu \quad \overset{k_{CH_3D+h\nu}}{\rightarrow} \quad products \tag{SR14}$$

**1.2 Reaction rate coefficients**

The reaction rates for the reaction (SR1–SR3) applied in this study are:

$$k_{CH_4+OH} \equiv 1.85 \times 10^{-20} \cdot \exp(2.82 \cdot \log(T) - \frac{987}{T}) \tag{1}$$

$$\equiv 1.85 \times 10^{-20} \cdot T^{2.82} \cdot \exp\left(-\frac{987}{T}\right) \tag{2}$$

$$k_{CH_4+Cl} \equiv 6.6 \times 10^{-12} \cdot \exp\left(\frac{-1240}{T}\right) \tag{3}$$

$$k_{CH_4+O1D} \equiv 1.75 \times 10^{-10} \tag{4}$$

Eq. (3) and (4) are from Sander et al. (2011) and Eq. (1) from Atkinson (2003). The temperature in [K] is denoted as $T$.

The reaction rate coefficients for the isotopologues (SR5–SR14) are achieved by multiplying the inverse of the corresponding Kinetic Isotope Effect (KIE) from Table 1 in the main manuscript. For example:

$$k_{^{13}CH_4+OH} \equiv k_{CH_4+OH} \cdot \text{KIE}^{OH-1}_{^{13}CH_4} \tag{5}$$

**2 Evaluation of simulated $CH_4$ isotopologues with observations**

The following section shows comparisons of simulation results with atmospheric observations from stationary surface sampling sites of the National Oceanic and Atmospheric Administration/Earth System Research Laboratory (NOAA/ESRL, White et al. (2016, 2017) with airborne observations taken during the Comprehensive Observation Network for TRace gases by AIrLiner (CONTRAIL) project (Umezawa et al., 2012), and with balloon borne observations by Röckmann et al. (2011). The study is based on work by Frank (2018) and observations are thereby compared to two simulations (1) EMAC-apos-02 and (2) EMAC-apos-03.

In the simulation EMAC-apos-02, the CH4 submodel together with its isotopologue extension is applied. This includes isotopologues concerning both, carbon, and hydrogen isotopes. The submodel is set up with the KIEs as introduced in Table 1 (see main manuscript). The comprehensive interactive chemistry simulation EMAC-apos-03 is conducted with the kinetic chemistry tagging hydrogen isotopologues, only. This configuration is chosen to investigate the pathways of deuterium from the source towards the end-product of deuterated methane ($CH_3D$), i.e. deuterated water vapour (HDO). This requires to include KIEs for the intermediates, too, as well as to apply adequate branching ratios and isotope transfer probabilities. The inclusion of carbon isotopologues with MECCA_TAG is omitted due to the fact that MECCA_TAG introduces additionally nearly twice as many chemical reactions and species as included in the basic chemical mechanism. To maintain a computational efficient simulation, the CH4 submodel is in EMAC-apos-03 additionally applied to simulate the carbon related methane ($CH_4$) isotopologues. In this case, the $CH_4$ tracer of the simplified $CH_4$ chemistry (CH4) submodel (`CH4_fx`), acting as the master tracer for the $CH_4$ isotopologues in the CH4 submodel, is in each model time step reset to the $CH_4$ tracer in the Module Efficiently Calculating the Chemi ensure an identical overall $CH_4$ budget. The CH4 submodel also uses directly the on-line calculated the hydroxyl radical (OH), excited oxygen (O($^1$D)) and chlorine (Cl) distribution from MECCA.

**Table  S1.**  The  isotopic  signature  of  the  emission  sources  as  used  in  the  model  simulations  with ECHAM/MESSy Atmospheric Chemistry (EMAC). All $\delta$-values and ranges are given in [permil (‰)].

| | $\delta^{13}C(CH_4)_{VPDB}$ | | | $\delta D(CH_4)_{VSMOW}$ | | |
|---|---|---|---|---|---|---|
| **Natural sources** | $\delta$-value | $\pm$ | references | $\delta$-value | $\pm$ | references |
| wetlands | -59.4 | 1.5 | [1,2,3,4,6] | -336.2 | 23.8 | [3,4,6] |
| other | | | | | | |
|    wildanimals | -61.5 | 0.5 | [1] | -319.0 | / | [5] |
|    termites | -63.3 | 6.5 | [1,2,3] | -390.0 | 35.5 | [3] |
|    volcanoes | -40.9 | 0.9 | [1,2] | -253.4 | 53.4 | [3,7] |
|    ocean (hydrates) | -59.0 | 1.0 | [1,2,3] | -220.0 | / | [3] |
| **Anthropogenic sources** | | | | | | |
| anthropogenic (collective) | -46.8 | 10.3 | [3,4,6,8] | -223.5 | 23.5 | [3,4,6] |
| rice | -63.0 | 1.0 | [1,2,3,4,6] | -324.3 | 5.5 | [3,4,6] |
| biomass burning | -23.9 | 1.6 | [1,2,3,4,6] | -213.0 | 7.5 | [3,4,6] |

references: [1] (Monteil et al., 2011) [2] (Fletcher et al., 2004) [3] (Whiticar and Schaefer, 2007) [4] (Snover and Quay, 2000) [5] (Rigby et al., 2012) [6] (Quay et al., 1999) [7] (Kiyosu, 1983) [8] (Zazzeri et al., 2015)

85    The applied emission inventory in the presented simulations is an a posteriori inventory derived using an inverse optimization technique (Frank, 2018; Bruhwiler et al., 2005). The specific isotopic signatures of the emission sources used in the model are listed in Table S1.

    The isotopic signatures are given in the $\delta$-notation (McKinney et al., 1950). We use the standard isotopic signature of Vienna Standard Mean Ocean Water (VSMOW) for the signature of D in $CH_4$ ($\delta D(CH_4)$) and Vienna-PeeDee Belemnite (VPDB) for 90  the signature of $^{13}C$ in $CH_4$ ($\delta^{13}C(CH_4)$).

**2.1  Surface sampling sites**

To start with the evaluation of the simulation results, isotopic observations from NOAA/ESRL sampling sites (White et al., 2016, 2017) are compared to the surface mixing ratios and $\delta$-values of the simulations. For the comparison a climatological mean of 2000–2009 is used, since this time period is represented by most of the stations and the dynamic equilibrium of the simulated isotopic 95  composition (as visible especially in EMAC-apos-03, Frank (2018)) has been reached.

    EMAC-apos-03 agrees well with the stations regarding the $CH_4$ mixing ratio. Interesting is that the $\delta^{13}C(CH_4)$ values are slightly better represented in EMAC-apos-02 compared to EMAC-apos-03, although the agreement is overall quite well in both simulations. This suggests that the emission signatures are a bit too low for methane containing $^{13}C$ ($^{13}CH_4$) in connection with the OH concentration in EMAC-apos-03. On the other hand, in case of $\delta D(CH_4)$, EMAC-apos-03 agrees better, however, is

[Figure]

**Figure S1.** Simulated multi-annual (2000–2009) surface mixing ratio of $CH_4$ in [nmoles of the chemical tracer per mole of air (mol mol$^{-1}$)] (upper), corresponding $\delta^{13}C(CH_4)_{VPDB}$ in [‰] (middle), and $\delta D(CH_4)_{VSMOW}$ in [‰] (lower). The left column shows results of EMAC-apos-02 and the right column those from EMAC-apos-03. The colored dots indicate the surface observations from NOAA/ESRL. The circles around the dots are the value of the simulation at the specific sampling height of the observation (in order to account for sub-grid orographic differences between simulation and observation).

[Figure]

**Figure S2.** Taylor diagrams of the comparison between observations and the simulations EMAC-apos-02 (blue) and EMAC-apos-03 (purple) at various surface sampling sites. The Taylor diagram is shown for $\delta^{13}C(CH_4)_{VPDB}$ (a) and for $\delta D(CH_4)_{VSMOW}$ (b) with respect to the representation of the annual cycle during the considered time period 2000–2009. The size of the triangles indicates the bias in percent with upward oriented triangles indicating a positive and downward oriented triangles a negative bias, respectively. Circles indicate a bias of less than 0.1%. The symbols below the diagram are stations outside the displayed range of the Taylor diagram and are indicated by the colored number. The normalized standard deviation is displayed by the upper black number and the correlation coefficient by the lower black number on the right hand side of the symbol.

still isotopically enriched compared to the station samples. This indicates that the chosen emission signatures for CH$_3$D are too heavy.

In addition to that, the annual cycle of the observations is generally fairly well represented in both simulations (see Fig. S2). However, the trend of the signatures at the stations over the years could not be captured yet. The reason for this is that the simulations fail to represent the general trend of the CH$_4$ mixing ratio and that the emission signatures of the individual sources are still uncertain.

**2.2 Airborne observations**

During the CONTRAIL project, atmospheric air samples were taken with an Automatic air Sampling Equipment (ASE) mounted on a commercial aircraft (Umezawa et al., 2012). These air samples were later measured concerning the isotopic composition of CH$_4$ using a gas chromatography system and a flame ionization detector. The here presented sampling data comprise several flights between 2006 and 2010, with each flight providing up to 12 air samples.

[Figure]

**Figure S3.** Observations provided by the CONTRAIL project (Umezawa et al., 2012). The green shaded area indicates region 1, and the red shaded area indicates region 2.

The presented flights are seperated into two regions, as depicted in Fig. S3. The first region (green) indicates the flights on a north-south route, bound from Narita airport (Japan) to Sydney, Brisbane (Australia) or Guam, and the second region (red) represents those flights on an east-west route, bound from Narita to Honolulu (Hawaii).

Especially the first region provides the opportunity to investigate the representation of the meridional gradient and the north-south imbalance in the $\delta$-values in the model as it nicely spans over the tropics ($40°$ S$-40°$ N). Simulation results and the airborne observations in this region are depicted in Fig. S4, where green dots indicate the observations. The dark green line indicates the mean of the observations and the shaded green area is the corresponding standard deviation. Simulated values are included as the red and blue dots respectively.

It is apparent from the shown results that the meridional gradient in the simulations concerning $CH_4$ and both isotopic signatures are well represented, although the absolute values differ. This indicates that the implemented KIE in the model is reasonable and that adjustments to the signatures of the emission inventory are required.

**2.3 Balloon borne observations**

The presented airborne observations are used to infer tropospheric chemical compositions. The high-altitude range of balloon borne observations enables to investigate the stratospheric isotopic signatures, as well.

The observational data are provided by Röckmann et al. (2011) and were obtained by altogether 13 balloon flights between 1987 and 2003 at four launch stations: Hyderabab in India (HYD), Aire sur l'Adour in France (ASA), Gap in France (GAP) and Kiruna in Sweden (KIR). The balloon-borne high-altitude air samples are obtained up to 10 hPa (35 km) and were later examined with respect to $CH_4$ mixing ratios as well as its isotopic composition concerning $^{13}CH_4$ and $CH_3D$ using a high precision continuous flow isotope ratio mass spectrometer (Brass and Röckmann, 2010).

The observations shown in Fig. S5 indicate two features:

[Figure]

**Figure S4.** Comparison of airborne observations (green) in the meridionally aligned region 1 with simulation data from EMAC-apos-02 (blue) and EMAC-apos-03 (red). $CH_4$ (a), $\delta^{13}C(CH_4)_{VPDB}$ (b) and $\delta D(CH_4)_{VSMOW}$ (c). The lighter red and blue colored markers indicate the de-biased simulation data for the direct comparison to the meridional gradient of the observations. The dark green line indicates the mean of the observations with the greenish shaded area being the corresponding single standard deviation.

- First, while $CH_4$ gets reduced towards higher altitudes, the isotopic content gets enriched (both, in $\delta^{13}C(CH_4)$ and $\delta D(CH_4)$). This occurs due to fractionation processes, which prefer lighter isotopologues in the sink reactions over heavier isotopologues.

- Secondly, again, a meridional gradient is visible. Polar regions tend to have less $CH_4$ than tropical regions, indicating to some extent the older age of the polar air masses. Consequently, polar regions are isotopically enriched compared to regions at mid and low latitudes.

The balloon-borne observations are compared to the simulations in Fig. S5 at pressure levels from 200 hPa to 10 hPa and separated into polar, mid-latitude and tropical regions. For the comparison, the monthly averaged data of the simulation is sampled at the specific year, month and location of the observation and interpolated from model levels to pressure levels. The plots in Fig. S5 further show the single standard deviation of the observations by the grey shaded areas and the standard deviation of all vertical profiles in the corresponding latitudinal region of the simulations as the shaded area in the color of the respective simulation.

The presented comparisons of observations to simulation results show that the global isotopic features of the meridional isotopic gradient and the isotopic gradient with altitude is captured well by both simulations (EMAC-apos-02, only with

[Figure]

**Figure S5.** Balloon borne observations from Röckmann et al. (2011, black) together with simulation results from EMAC-apos-02 (blue) and EMAC-apos-03 (red). The rows of panels from top to bottom present balloon launches in the polar region from Kiruna in Sweden (KIR), in the mid-latitude region from Aire sur l'Adour in France (ASA) and Gap in France (GAP), and in the tropical region from Hyderabab in India (HYD). The profiles of the simulations are taken from monthly averaged data in the specific year, month and at the location of the observation. For observations taken before the simulation start, the simulated year 1990 is shown. Shaded areas indicate the single standard deviation of the observations (grey) and the simulations (blue and red, respectively) with respect to the variations within the specific latitudinal region and the interannual variation in the years 1990–2003.

145    CH4 submodel, and EMAC-apos-03, with MECCA and the CH4 submodel). This indicates that the implementation of the simulation of $CH_4$ isotopologues is sufficiently realized and also confirms the suitability of the chosen KIE values. Absolute values and the inter-annual trend of the observations, however, are not captured well, which is mainly caused by uncertainties in the $CH_4$ emission fluxes and the applied source signatures.

[revised manuscript text omitted]
. **217 -59.4 1.5** [1,2,3,4,6] **-336.2 23.8** [3,4,6] **126 -50.3 8.9 -313.3 88.9** 40 -53.8 /[3] -385.0 /[3] 15 -61.5 0.5[1] -319.0 /[5] 11 -63.3 6.5[1,2,3] -390.0 35.5 [3] 54 -40.9 0.9[1,2] -253.4 53.4 [3,7] 6 -59.0 1.0[1,2,3] -220.0 /[3] **200 -57.5 3.8 -313.8 26.5** 89 -60.2 0.3[3,4,6] -317.5 12.5 [3,4] 75 -51.7 2.5[3,4,6] -304.3 8.5[3,4,6] 36 -63.0 1.0[1,2,3,4,6] -324.3 5.5[3,4,6] **96 -41.8 7.5 -154.2 2.5** 32 -43.5 0.5[3,6] -182.5 2.5[3,6] 64 -41.0 7.0[3,6,8] -140.0 0.0[3,6] **35 -23.9 1.6** [1,2,3,4,6] **-213.0 7.5** [3,4,6] -59.0 -324.5 -41.8 -192.0 -23.9 -213.0~~

[revised manuscript text omitted]